# Reinforcement Learning in Reward-Mixing MDPs

**Jeongyeol Kwon**
The University of Texas at Austin
kwonchungli@utexas.edu

**Yonathan Efroni**
Microsoft Research, NYC
jonathan.efroni@gmail.com

**Constantine Caramanis**
The University of Texas at Austin
constantine@utexas.edu

**Shie Mannor**
Technion, NVIDIA
shie@ee.technion.ac.il,
smannor@nvidia.com

## Abstract

Learning a near optimal policy in a partially observable system remains an elusive challenge in contemporary reinforcement learning. In this work, we consider episodic reinforcement learning in a reward-mixing Markov decision process (MDP). There, a reward function is drawn from one of multiple possible reward models at the beginning of every episode, but the identity of the chosen reward model is not revealed to the agent. Hence, the latent state space, for which the dynamics are Markovian, is not given to the agent. We study the problem of learning a near optimal policy for two reward-mixing MDPs. Unlike existing approaches that rely on strong assumptions on the dynamics, we make no assumptions and study the problem in full generality. Indeed, with no further assumptions, even for two switching reward-models, the problem requires several new ideas beyond existing algorithmic and analysis techniques for efficient exploration. We provide the first polynomial-time algorithm that finds an $\epsilon$-optimal policy after exploring $\tilde{O}(poly(H, \epsilon^{-1}) \cdot S^2 A^2)$ episodes, where $H$ is time-horizon and $S$, $A$ are the number of states and actions respectively. This is the first efficient algorithm that does not require any assumptions in partially observed environments where the observation space is smaller than the latent state space.

## 1 Introduction

In reinforcement learning (RL), an agent solves a sequential decision-making problem in an unknown dynamic environment to maximize the long-term reward [46]. The agent interacts with the environment by receiving feedback on its actions in the form of a state-dependent reward and observation.

One of the key challenges in RL is exploration: in the absence of auto-exploratory properties such as ergodicity of the system, the agent has to devise a clever scheme to collect data from under-explored parts of the system. In a long line of work, efficient exploration in RL has been extensively studied for fully observable environments, *i.e.,* under the framework of Markov decision process (MDP) with a number of polynomial-time algorithms proposed [34, 27, 43, 40, 18]. For instance, in tabular MDPs, *i.e.,* environments with a finite number of states and actions, we can achieve sample-optimality or minimax regret without any assumptions on system dynamics [3, 50].

In contrast to fully observable environments, very little is known about the exploration in partially observable MDPs (POMDPs). In general, RL in POMDPs may require an exponential number of samples (without simplifying structural assumptions) [35, 31]. Therefore, it is important to consider natural sub-classes of POMDPs which admit tractable solutions. Previous studies focus on a special class of POMDPs where observation spaces are large enough such that a single observation provides

sufficient information on the underlying state [22, 35, 4, 12, 16, 30]. However, there are many applications in robotics, topic modeling and beyond, where the observations space is smaller than the latent state space. This "small observation space" setting is critical, as most prior work does not apply. Our work seeks to provide one of the first results to tackle a class of problems in this space.

In particular, we tackle a sub-class of POMDPs inspired by latent variable or mixture problems. Specifically, we consider reward-mixing MDPs (RM-MDPs). In RM-MDPs, the transition kernel and initial state distribution are defined as in standard MDPs, while the reward function is randomly chosen from one of $M$-reward models at the beginning of every episode (see the formal definition 1). RM-MDPs are important in their own right. Consider, for instance, a user-interaction model in a dynamical web system, where reward models vary across users with different characteristics [38, 23]. A similar setting has been considered in a number of recent papers [8, 6, 23, 45, 7], and most recently, and most directly related to our setting, [36].

Even for $M = 2$, the RM-MDP setting shares some of the key challenges of general POMDPs. This is because the best policy for the RM-MDP problem should account for not just the current state, but also an entire sequence of previous observations since some previous events might be correlated with a reward of the current action. And in the small observation space setting, prior work as in [22, 35, 4, 12, 16, 30] cannot be applied. The work in [36] develops an algorithmic framework for solving the RM-MDP problem in this small observation setting, but requires strong assumptions without which the algorithm falls apart. We discuss this in more detail below. Indeed, no work to date, has been able to address the RM-MDP problem, even for $M = 2$, without significant further assumptions. This is precisely the problem we tackle in this paper.

**Main Results and Contributions**    We focus on the problem of learning near-optimal policies in two reward-mixing MDPs, *i.e.,* RM-MDPs with two reward models $M = 2$. To the best of our knowledge, no prior work has studied sample complexity of RL in RM-MDPs even for $M = 2$ without any assumptions on system dynamics. Specifically, we provide the first polynomial-time algorithm which learns an $\epsilon$-optimal policy after exploring $\tilde{O}(poly(H, \epsilon^{-1}) \cdot S^2 A^2)$ episodes. We also show that $\Omega(S^2 A^2/\epsilon^2)$ number of episodes is necessary, and thus our result is tight in $S$, $A$. This is the first efficient algorithm in a sub-class of partially observable domains with small observation spaces and without any assumptions on system dynamics.

On the technical side, we must overcome several new challenges that are not present in fully observable settings. One key hurdle relates to identifiability. In standard MDPs, an average of single observations from a single state-action pair is sufficient to get unbiased estimators of transition and reward models for the state-action. However, in RM-MDPs, the same quantity would only give an *averaged reward* model, where the average is taken over the reward model distribution. However, an average reward model alone is not sufficient to obtain the optimal policy which depends on a sequence of previous rewards.

Our technique appeals to the idea of *uncertainty in higher-order moments*, bringing inspiration from algorithms for learning a mixture of structured distributions [19, 9, 26, 15]. In such problems, the key for efficient learning is to leverage information from higher-order moments. Following this spirit, we consider estimating correlations of rewards at multiple different state-actions. A central challenge we must overcome is that in finite-horizon MDPs, it may be not possible to estimate all higher-order correlations with uniformly good accuracy. In fact, it may not be possible to estimate some correlations at all when some pairs of state-actions are not reachable in the same episode. Therefore, we cannot simply rely on existing techniques that require good estimations of *all* elements in higher-order moment matrices. The main technical challenge is, therefore, to show that a near-optimal policy can still be obtained from *uncertain and partial correlations* of state-actions. Our technical contributions are thus two-fold: (1) design of an efficient exploration algorithm to estimate each correlation up to some required accuracy, and (2) new technical tools to analyze errors from different levels of uncertainties in estimated higher-order correlations.

**Related Work**    Efficient exploration in fully observed environments has been extensively studied in the framework of MDPs. In tabular MDPs, a long line of work has studied efficient exploration algorithms and their sample complexity based on the principle of optimism [27, 43, 3, 18, 50, 47, 44] or posterior-sampling [1, 43, 40, 41]. Beyond tabular settings, recent work seeks for efficient exploration in more challenging settings such as with large or continuous state spaces [28, 32, 17].

All aforementioned work considers only fully observable environments where the complexity of exploration is significantly lower as there is no need to consider observation histories.

Most previous study on exploration in POMDPs focuses on large observation spaces such that a set of single observations is statistically sufficient to infer internal states, *i.e.,* the observation matrix has non-zero minimum singular value. Under such environments, previous work either considers a strong assumption on the uniform ergodicity of the latent state transitions [25, 5, 22, 4] or the uniqueness of underlying states for every observation [35, 12, 16]. The exception is a recent work by Jin et al. [30] which studied efficient exploration without any assumptions on latent system dynamics.

One of the most closely related to problem to our work is a reinforcement learning problem with latent contexts (also referred as multitask RL), which is considered in [6, 23, 36]. Previous approaches relied on several strong assumptions such as very long time horizon [6, 23], revealed contexts in hindsight or ergodic properties of system dynamics [36]. Our work takes a first step without such assumptions by focusing on environments where only the reward model differs across different tasks.

RM-MDP might be also viewed as a special case of adversarial MDPs (*e.g.,* [49, 39, 29]) with an oblivious adversary who can only play within a finite set of reward models. However, our goal is to find an optimal policy in a broader class of *history-dependent* policies, whereas in adversarial MDP literature they compare only to the best Markovian policy in hindsight. RM-MDP can be also considered as an instance of latent variable models (*e.g.,* [14, 20, 13, 25, 24]), where sample reward sequences are influenced by unobserved confounders. RM-MDP poses several new challenges as both challenges from reinforcement learning and statistical inference are interlaced by latent contexts.

## 2 Problem Setup

We define the problem of episodic reinforcement learning problem with time-horizon $H$ in reward-mixing Markov decision processes (RM-MDPs) defined as follows:

**Definition 1 (Reward-Mixing Markov Decision Process (RM-MDP))** *Let* $\mathcal{M}$ *be a tuple* $(\mathcal{S}, \mathcal{A}, T, \nu, \{w_m\}_{m=1}^M, \{R_m\}_{m=1}^M)$ *be a RM-MDP on a state space* $\mathcal{S}$ *and action space* $\mathcal{A}$ *where* $T : \mathcal{S} \times \mathcal{A} \times \mathcal{S} \to [0, 1]$ *is a common transition probability measures that maps a state-action pair and a next state to a probability, and* $\nu$ *is a common initial state distribution. Let* $S = |\mathcal{S}|$ *and* $A = |\mathcal{A}|$. $w_1, ..., w_M$ *are the mixing weights such that at the beginning of every episode, one reward model* $R_m$ *is randomly chosen with probability* $w_m$. *A reward model* $R_m : \mathcal{S} \times \mathcal{A} \times \{0, 1\} \to [0, 1]$ *is a probability measure for rewards that maps a state-action pair and a binary reward to a probability for* $m \in [M]$.

In this work, we focus on a special case of RM-MDPs when $M = 2$, *i.e.,* 2RM-MDPs. For the ease of presentation, we assume all random rewards take values from $\{0, 1\}$, *i.e.,* rewards are binary random variables. In this work, we assume a uniform-prior over each latent context such that $w_1 = w$ and $w_2 = 1 - w$ with $w = 1/2$. Detailed discussions on these simplifying assumptions can be found in Section 5. We consider a policy class $\Pi$ which contains all history-dependent policies $\pi : (\mathcal{S}, \mathcal{A}, \{0, 1\})^* \times \mathcal{S} \to \mathcal{A}$. The goal of the problem is to find a near optimal policy $\pi \in \Pi$ that has near optimal value w.r.t. the optimal one, $V_{\mathcal{M}}^* := \max_{\pi \in \Pi} \sum_{m=1}^M w_m \mathbb{E}_m^\pi \left[ \sum_{t=1}^H r_t \right]$, where $\mathbb{E}_m^\pi[\cdot]$ is expectation taken over the $m^{th}$ MDP with a policy $\pi$. More formally, we wish our algorithm to return an $(\epsilon, \eta)$ provably-approximately-correct (PAC) optimal policy, which we also refer as a near optimal policy, defined as follows:

**Definition 2 (($\epsilon, \eta$)-PAC optimal policy)** *An algorithm is* $(\epsilon, \eta)$*-PAC if it returns a policy* $\hat{\pi}$ *s.t.*

$$\mathbb{P}(V_{\mathcal{M}}^* - V_{\mathcal{M}}^{\hat{\pi}} \le \epsilon) \ge 1 - \eta.$$

### 2.1 Notation

We often denote a state-action pair $(s, a)$ as one symbol $x = (s, a) \in \mathcal{S} \times \mathcal{A}$. For any sequence $(y_1, y_2, ..., y_t)$ of length $t$, we often simplify the notation as $(y)_{1:t}$ for any symbol $y$. We denote $l_1$ sum of any function $f$ over a random variable $X$ conditioned on an event $E$ as $\|f(X|E)\|_1 = \sum_{X \in \mathcal{X}} |f(X|E)|$, where $\mathcal{X}$ is a support of $X$. $\mathbb{1}\{\mathcal{E}\}$ is an indicator function for any event $\mathcal{E}$ and

---

**Algorithm 1** Learning Two Reward-Mixture MDPs

---

1: Run pure-exploration (Algorithm 2) to estimate second-order correlations
2: Estimate 2RM-MDP parameters $\hat{\mathcal{M}}$ from the collected data (Algorithm 3)
3: Return $\hat{\pi}$, the (approximately) optimal policy of $\hat{\mathcal{M}}$

---

$\mathbb{P}(\mathcal{E})$ is a probability of any event $\mathcal{E}$ measured without contexts. If we use $\mathbb{P}$ without any subscript, it is a probability of an event measured outside of the context, *i.e.*, $\mathbb{P}(\cdot) = \sum_{m=1}^{M} w_m \mathbb{P}_m(\cdot)$. We refer $\mathbb{P}_m$ the probability of any event measured in the $m^{th}$ context (or in $m^{th}$ MDP). If probability of an event depends on a policy $\pi$, we add superscript $\pi$ to $\mathbb{P}$. We denote $V_{\mathcal{M}}^{\pi}$ as an expected long-term reward for model $\mathcal{M}$ with policy $\pi$. We use $\hat{\cdot}$ to denote empirical counterparts. For any set $\mathcal{A}$ and $d \in \mathbb{N}$, $\mathcal{A}^{\otimes d}$ is a $d$-ary Cartesian product over $\mathcal{A}$. We use $a \vee b$ to refer $\max(a, b)$ and $a \wedge b$ to mean $\min(a, b)$ for $a, b \in \mathbb{R}$.

Specifically for $M = 2$: We use shorthand $p_m(x) := R_m(r = 1|x)$ for $m = 1, 2$. Let an expected averaged reward $p_+(x) := \frac{1}{2}(p_1(x) + p_2(x))$, differences in rewards $p_-(x) := \frac{1}{2}(p_1(x) - p_2(x))$ and $\Delta(x) := |p_-(x)|$.

## 3 Algorithms

Before developing a learning algorithm for 2RM-MDP, let us provide intuition behind our algorithm. At a high-level, our approach lies on the following observation:

*The latent reward model of 2RM-MDP can be recovered from reward* correlations *and the* averaged *reward*.

Consider a simpler interaction model, in which we have a perfect and exact access to correlations of the reward function. That is, suppose we can query for any $(x_i, x_j) \in (\mathcal{S} \times \mathcal{A})^{\otimes 2}$ the reward correlation function

$$\mu(x_i, x_j) := \mathbb{E}\left[r_i \cdot r_j | x_i, x_j\right] = \frac{1}{2} \cdot (p_1(x_i)p_1(x_j) + p_2(x_i)p_2(x_j)). \tag{1}$$

Assume we can also query the exact expected average reward $p_+(x)$ for all $x \in \mathcal{S} \times \mathcal{A}$. Suppose we can construct a matrix $B \in \mathbb{R}^{SA \times SA}$ indexed by state-actions $x \in \mathcal{S} \times \mathcal{A}$, such that at its $(i, j)$ entry is given by:

$$B_{i,j} = \mu(x_i, x_j) - p_+(x_i)p_+(x_j).$$

Simple algebra shows that $B = qq^{\top}$ where $q \in \mathbb{R}^{SA}$ is a vector indexed by $x \in \mathcal{S} \times \mathcal{A}$ such that $q_i = p_-(x_i)$. Hence, we can compute the top principal component of $B$, from which we can recover $p_-(x)$ for all $x \in \mathcal{S} \times \mathcal{A}$. Through the access to $p_-(x)$ (ignore sign ambiguity issue for now) and $p_+(x)$ we can recover the probabilities conditioned on the latent contexts via

$$R_1(r = 1|x) = p_+(x) + p_-(x) \ \text{and} \ R_2(r = 1|x) = p_+(x) - p_-(x). \tag{2}$$

Once we have the latent reward model we can use any approximate planning algorithm (e.g., point-based value iteration) to find an $\epsilon$-optimal policy.

However, when such exact oracle is not supplied, and the interaction with the 2RM-MDP is based on trajectories and roll-in policies, getting a point-wise good estimate of $B$ is not generally possible. For example, some state-action pairs cannot be visited in the same episode, in which case we cannot get any samples for the correlations of such pairs of state-actions: $\mu(x_i, x_j)$ is completely unknown. In such a case, recovery of the true model parameters is not possible in general even if we estimate all reachable pairs *without errors*. Hence the main challenge is as follows: can we estimate a model from a set of empirical second-order correlations such that the estimated model is close to the true model in terms of the optimal policy, even if it is not close in model parameters?

For usual MDPs, states that cannot be reached pose no problems: we do not need to learn transition or reward probabilities of these states, since no optimal policy will use that state. Indeed, with this observation at hand, *reward free exploration* techniques [31, 33] can be utilized to learn a good model w.r.t. all possible reward functions (given a fixed initial distribution). Our main conceptual contribution is to show how these techniques can be leveraged for the RM-MDP problem.

**Overview of our approach.** Our approach develops the following ideas.

1. Section 3.1: We define what we call a *Second Order MDP*, an augmented MDP whose states are pairs of states of the original MDP. Borrowing technology from single MDPs, we show that reward free exploration on the augmented MDP can be used to approximate second order moments of the original MDP, along with confidence intervals, obtained by how often a particular pair of states can be reached.

2. Section 3.2: We show how to find the parameters of a model whose second order moments are in the confidence set obtained from the augmented MDP. To do this, we show that we can use linear programming to select an element of the uncertainty set that corresponds to valid second order moments for a model.

3. Section 3.3: The main contribution of the analysis, is then to show that for any model whose true second order statistics lie in this set obtained, an approximate planning oracle is guaranteed to return an $\epsilon$-optimal solution to the true model. Theorem 3.2 shows that sample complexity depends quadratically in $(S \cdot A)$. This is well-expected, given that our algorithm must compute statistics on the augmented graph. We next give a matching lower bound: Theorem 3.3 shows that this quadratic dependence is unavoidable.

## 3.1 Pure Exploration of Second-Order Correlations

Our first goal is to collect samples for *pairs* of state-actions through exploration. This will ultimately allow us to estimate the correlations of the reward function, as well as the average reward per-state $p_+(x)$. The backbone of pure-exploration is adaptive reward-free exploration schemes studied in standard MDP settings [33]. We first consider a surrogate model that helps to formulate data collection process for correlations of pairs. Specifically, we first define the augmented MDP:

**Definition 3 (Augmented Second-Order MDPs)** *An augmented second-order MDP $\widetilde{\mathcal{M}}$ is defined on a state-space $\widetilde{\mathcal{S}}$ and action-space $\widetilde{\mathcal{A}}$ where*

$$\widetilde{\mathcal{S}} = \left\{ (i, v, s) \mid i \in \{1, 2, 3\}, v : (v^1, v^2) \in ((\mathcal{S} \times \mathcal{A}) \cup \{null\})^{\otimes 2}, s \in \mathcal{S} \right\},$$

$$\widetilde{\mathcal{A}} = \{(a, z) \mid a \in \mathcal{A}, z \in \{0, 1\}\}.$$

*Then in $\widetilde{\mathcal{M}}$, an augmented state $(i_t, v_t, s_t)$ evolves under an action $(a_t, z_t)$ as follows:*

$$i_1 = 1, v_1 = (null, null), s_1 \sim \nu(\cdot), \quad s_{t+1} \sim T(\cdot|s_t, a_t), \quad v_{t+1}^j = v_t^j \ \ \forall j \in [2]/\{i_t\},$$

$$i_{t+1} = \begin{cases} i_t & \text{if } z_t = 0 \text{ or } i_t = 3 \\ i_t + 1 & \text{else} \end{cases}, \quad v_{t+1}^{i_t} = \begin{cases} v_t^{i_t} & \text{if } z_t = 0 \text{ or } i_t = 3 \\ (s_t, a_t) & \text{else} \end{cases}. \tag{3}$$

The second-order augmented MDP consists of an extended state space where the first two coordinates $i, v$ store previously selected state-actions followed by the coordinate $s$ for the current state. The action space is a product of original actions $a$ and choice actions $z$, where $z = 1$ means that we select the current state-action to be explored as a part of a pair in the episode. Specifically, when $z = 1$, by design we increase the count $i$. If $i$ reaches 3, it means we collected a sample of a pair $(v^1, v^2)$ in the episode.

Initially the true model parameters are unknown and thus we are not aware of how to collect samples for any pair of state-actions or how much samples we need for each pair. In order to resolve this, we use ideas from reward free exploration for MDPs: we consider the upper confidence error bound function $\widetilde{Q}$ that follows from the Bellman-equation for $\widetilde{\mathcal{M}}$ with (pure) exploration bonus:

$$b_r(i, v, z) = \mathbb{1}\{i = 2 \cap z = 1\} \cdot \left(1 \wedge \sqrt{\frac{\iota_2}{n(v')}}\right), \ b_T(s, a) = \left(1 \wedge \sqrt{\frac{\iota_T}{n(s, a)}}\right),$$

$$\widetilde{Q}_t((i, v, s), (a, z)) = 1 \wedge \left(b_r(i, v, z) + \mathbb{E}_{s' \sim \hat{T}(\cdot|s, a)}\left[\widetilde{V}_{t+1}(i', v', s')\right] + b_T(s, a)\right), \tag{4}$$

where $\mathbb{1}\{i = 2 \cap z = 1\}$ is an indicator of whether to collect samples for correlations between state-actions in $v'$. Here, $i'$ and $v'$ are first and second coordinates of the next state following the transition rule (3), $\iota_2 = O(\log(K/\eta))$, $\iota_T = O(S \log(K/\eta))$ are properly set confidence interval parameters,

**Algorithm 2** Pure Exploration of Second-Order Correlations

---

1: Initialize $\widetilde{Q}_{(\cdot)}(\cdot) = \widetilde{V}_0 = 1$, $n(v) = 0$ for all $v \in (\mathcal{S} \times \mathcal{A})^{\otimes 2}$.
2: **while** $\widetilde{V}_0 > \epsilon_{pe}$ **do**
3:     Get an initial state $s_1$ for the $k^{th}$ episode. Let $v_1 = (null, null)$, $i = 1$, $r_c = 1$
4:     **for** $t = 1, 2, ..., H$ **do**
5:         Pick $(a_t, z_t) = arg\max_{(a,z) \in \widetilde{\mathcal{A}}} \widetilde{Q}_t((i_t, v_t, s_t), (a, z))$.
6:         Play action $a_t$, observe next state $s_{t+1}$ and reward $r_t$.
7:         **if** $i_t \leq 2$ **and** $z = 1$ **then**
8:             $r_c \leftarrow r_c \cdot r_t$
9:         **end if**
10:        Update $(i_{t+1}, v_{t+1}, s_{t+1})$ according to the choice of $a_t$ and $z_t$ following the rule in (3)
11:     **end for**
12:     **if** $i_{H+1} = 3$ **then**
13:         $n(v_{H+1}) \leftarrow n(v_{H+1}) + 1$,    $\hat{\mu}(v_{H+1}) \leftarrow (1 - 1/n(v_{H+1}))\,\hat{\mu}(v_{H+1}) + r_c$
14:     **end if**
15:     Update $\hat{\nu}, \hat{T}, \hat{p}_+$ from the trajectory $(s, a, r)_{1:H}$, and then update $\widetilde{Q}$ and $\widetilde{V}$ using (4), (5)
16: **end while**

---

$K$ is the total number of episodes to be explored and $\widetilde{Q}_{H+1}(\cdot) = 0$. At every episode, we choose a greedy action with respect to $\widetilde{Q}$ and collect sufficient data for pairs of state-actions by pure exploration on the augmented MDP. We repeat the pure-exploration process until $\widetilde{V}_0$ becomes less than the threshold $\epsilon_{pe}$ where

$$\widetilde{V}_t(i, v, s) = \max_{(a', z') \in \widetilde{\mathcal{A}}} \widetilde{Q}_t((i, v, s), (a', z')), \quad \widetilde{V}_0 = \sqrt{\iota_\nu/K} + \sum_s \hat{\nu}(s) \cdot \widetilde{V}_1(1, v_1, s), \quad (5)$$

and $\iota_\nu = O(S \log(K/\eta))$ is a confidence interval parameter for initial states. The pure-exploration procedure is summarized in Algorithm 2. The main purpose of Algorithm 2 is to balance the amount of samples for correlations.

## 3.2 Recovery of Empirical Model with Uncertainty

Once we gather enough samples to estimate correlations of pairs of state-actions we are assured that $\tilde{V}_0 \leq \epsilon_{pe}$. Given this, we describe an efficient algorithm to recover a good estimate of the true 2RM-MDP parameters. We formulate the model recovery problem as a linear program (LP). Recall that for any $x_i, x_j \in (\mathcal{S} \times \mathcal{A})^{\otimes 2}$, we can compute the multiplication of differences at two positions $x_i, x_j$:

$$u(x_i, x_j) := p_-(x_i)p_-(x_j) = \mu(x_i, x_j) - p_+(x_i)p_+(x_j). \quad (6)$$

Ideally with exact oracles for getting values of $u(x_i, x_j)$, we can recover $p_-(x)$ through LP: let $l(x) := \log |p_-(x)|$ for all $x \in \mathcal{S} \times \mathcal{A}$. Then we first find a solution for the following LP:

$$l(x_i) + l(x_j) = \log |\mu(x_i, x_j) - p_+(x_i)p_+(x_j)|, \; \forall x_i, x_j \in \mathcal{S} \times \mathcal{A}.$$

After solving for $l(x)$, we can find consistent assignments of signs $sign(x)$ such that $p_-(x) = sign(x) \cdot \exp(l(x))$, which can be formulated as 2-Satisfiability problem [2].

However,, after the pure-exploration phase, we only have estimates of correlations $\hat{\mu}(x_i, x_j)$ and confidence intervals $b(x_i, x_j) := \sqrt{\iota_2/n(x_i, x_j)} + \epsilon_0^2$ where $\epsilon_0 = \epsilon/H^2$ is a small tolerance parameter (as well as estimates of the expected average reward $\hat{p}_+(x)$ and its confidence interval $b(x) := \sqrt{\iota_2/n(x)}$). Note that $n(x_i, x_j)$ is the number of counts that a pair $(x_i, x_j)$ is counted during pure-exploration phase. Let $\hat{u}(x_i, x_j) := \hat{p}_-(x_i)\hat{p}_-(x_j)$ be an empirical estimate of $u(x_i, x_j)$. We want to recover $\hat{p}_-(x) = sign(x) \cdot \exp(\hat{l}(x))$ such that the following is satisfied:

$$|(\hat{\mu}(x_i, x_j) + \hat{p}_+(x_i)\hat{p}_+(x_j)) - \hat{u}(x_i, x_j)| \leq b(x_i, x_j), \; \forall x_i, x_j \in \mathcal{S} \times \mathcal{A}. \quad (7)$$

We present in Appendix B.1 the LP formulation to recover $\hat{l}(x)$ and $sign(x)$. The outline of the procedure is stated in Algorithm 3 and more details are presented in Algorithm 4. In order for Algorithm 3 to succeed, we need the following lemma on the existence of feasible solutions in the LP with high probability:

**Algorithm 3** Reward Model Recovery from Second-Order Correlations

1: Solve an LP for $\hat{l}(x)$ and find $sign(x)$ for all $x \in \mathcal{S} \times \mathcal{A}$ that satisfies (7).
2: Clip $\hat{l}(x)$ within $(-\infty, \hat{u}(x)]$ where $\hat{u}(x) = \log\left(\min(\hat{p}_+(x), 1 - \hat{p}_+(x))\right)$ for all $x \in \mathcal{S} \times \mathcal{A}$.
3: Set $\hat{p}_-(x) = sign(x) \cdot \exp(\hat{l}(x))$ for all $x \in \mathcal{S} \times \mathcal{A}$.
4: Let $\hat{p}_1(x) = \hat{p}_+(x) + \hat{p}_-(x), \hat{p}_2(x) = \hat{p}_+(x) - \hat{p}_-(x)$ for all $x \in \mathcal{S} \times \mathcal{A}$.
5: Return $\hat{\mathcal{M}} = (\mathcal{S}, \mathcal{A}, \hat{T}, \hat{\nu}, \{\hat{R}_m\}_{m=1}^2)$, an empirical 2RM-MDP model.

**Lemma 3.1** *Suppose that we set the confidence parameters as $\iota_1 = \iota_2 = O(\log(SA/\eta))$ and $\iota_\nu = \iota_T = O(S\log(SA/\eta))$. Then with probability at least $1 - \eta$, Algorithm 3 returns a model $\hat{\mathcal{M}}$ that satisfies constraints (7) for all $x_i, x_j \in \mathcal{S} \times \mathcal{A}$.*

Given Lemma 3.1, Algorithm 3 is able to find at least one feasible solution, i.e., model $\hat{\mathcal{M}}$, in polynomial-time. Via such a solution, we obtain $\hat{p}_- \in \mathbb{R}^{SA}$ and can recover an estimate of the latent model $p_1(s, a) = \hat{R}_1(r = 1|s, a)$ and $p_2(s, a) = \hat{R}_2(r = 1|s, a)$ for all state-action pairs by a plug-in estimate (2). We refer to such a model as the *empirical model*. The proof of Lemma 3.1 is given in Appendix B.2.

### 3.3 Main Theoretical Results

**Upper Bound** We first prove the correctness of our main algorithm (Algorithm 1) and compute an upper bound on its sample complexity. Our upper bound has a quadratic dependence on $S \cdot A$. This is well-expected because we must estimate quantities related to the augmented second order MDP. Our lower bound shows that this dependence is in fact tight. Additionally, our bounds exhibit an interesting dependence on the error threshold, $\epsilon$, that in fact depends on the minimum separation in distinguishable state-actions:

$$\delta = 1 \wedge \min_{x \in \mathcal{S} \times \mathcal{A}: \Delta(x) > 0} \Delta(x). \tag{8}$$

where $\Delta(x) := |p_-(x)|$. This is reminiscent of similar results for mixture models (e.g., [10, 37]). We state the following theorem on the sample-complexity of 2RM-MDPs.

**Theorem 3.2** *There exists a universal constant $C > 0$ such that if we run Algorithm 1 with $\epsilon_{pe} = \frac{\epsilon}{H^3} \cdot \left(\delta \vee \frac{\epsilon}{H^2}\right)$, then Algorithm 2 terminates after at most $K$ episodes where,*

$$K = C \cdot H^6 \cdot \frac{S^2 A}{\epsilon^2} \cdot (H + A) \cdot \left(\frac{\epsilon}{H^2} \vee \delta\right)^{-2} \cdot \log(HSA/\epsilon\eta) \cdot \log^2(H/\epsilon),$$

*with probability at least $1 - \eta$. Furthermore, under the same high probabilistic event, Algorithm 1 returns an $\epsilon$-optimal policy.*

Theorem 3.2 shows that we can find a $\epsilon$-optimal policy for any 2RM-MDP instance with at most $O\left(poly(H, \epsilon^{-1}) \cdot S^2 A^2\right)$ episodes of exploration. We leave it as future work to investigate whether dependency on $\epsilon$ can be uniformly $O(1/\epsilon^2)$ regardless of $\delta$, as well as whether the dependency on $H$ can be tightened.

**Lower Bound** In Theorem 3.2, the sample complexity guaranteed by Algorithm 1 is $\Omega(S^2 A^2)$. We show that the dependency on $S^2 A^2$ is also necessary for two reward-mixing MDPs:

**Theorem 3.3** *There exists a class of 2RM-MDPs such that in order to obtain $\epsilon$-optimal policy, we need at least $\Omega(S^2 A^2/\epsilon^2)$ episodes.*

The proof of the lower bound can be found in Appendix C.

**Computational Complexity**   Main bottlenecks of Algorithm 1 are in two parts: (1) solving the LP in Algorithm 3 and (2) computing the optimal policy for $\hat{\mathcal{M}}$. For (1), with $SA$ variables and $O(S^2A^2)$ constraints, solving the LP in Algorithm 3 takes $poly(S, A)$ times (see [11] and the references therein for some known computational complexity results for solving LPs). For (2), point-based value-iteration algorithm [42] runs in $O(\epsilon^{-2}H^5SA)$ time to obtain $\epsilon$-optimal policy for any specified 2RM-MDP model. Therefore, the overall running time of Algorithm 1 is polynomial in all parameters.

## 4   Analysis Overview

For the ease of presentation, we assume that the true transition and initial state probabilities are known in advance. In Appendix B, we complete the proof of Theorem 1 without assuming known transition and initial probabilities but instead using $\hat{\nu}$ and $\hat{T}$ estimated during the pure-exploration phase.

We first note that given any policy $\pi \in \Pi$, difference in expected rewards can be bounded by $l_1$-statistical distance in trajectory distributions. More specifically, consider the set of all possible trajectories $\mathcal{T} = (\mathcal{S} \times \mathcal{A} \times \{0, 1\})^{\otimes H}$, that is, any state-action-reward sequence of length $H$. Then,

$$|V_{\mathcal{M}}^\pi - V_{\hat{\mathcal{M}}}^\pi| \leq H \cdot \|(\mathbb{P}^\pi - \hat{\mathbb{P}}^\pi)((x, r)_{1:H})\|_1 = \sum_{\tau \in \mathcal{T}} |\mathbb{P}^\pi(\tau) - \hat{\mathbb{P}}^\pi(\tau)|,$$

where $\hat{\mathbb{P}}^\pi(\cdot)$ is the probability of an event measured in the empirical model $\hat{\mathcal{M}}$ with policy $\pi$. Therefore, for any policy $\pi \in \Pi$, our goal is to show that $\sum_{\tau \in \mathcal{T}} |\mathbb{P}^\pi(\tau) - \hat{\mathbb{P}}^\pi(\tau)| \leq O(\epsilon/H)$, *i.e.,* the true and empirical models are close in $l_1$-statistical distance in trajectory distributions.

The main challenge in the analysis is to bound the $l_1$ distance when estimated first and second-order moments are within the range of confidence intervals. We now show that when all pairs of state-actions in $\tau$ were visited sufficient amount of times then we can bound the $l_1$ difference. Let us first divide a set of trajectories into several groups depending on the number of times that every pair in each trajectory has been explored during the pure-exploration phase. We first define the following sets:

$$\mathcal{X}_l = \{(x_i, x_j) \in (\mathcal{S} \times \mathcal{A})^{\otimes 2} \mid n(x_i, x_j) \geq n_l\},$$
$$\mathcal{E}_l = \{x_{1:H} \in \mathcal{T} \mid \forall t_1, t_2 \in [H],\ t_1 \neq t_2\ \text{s.t.}\ (x_{t_1}, x_{t_2}) \in \mathcal{X}_l\}, \tag{9}$$

where $n_l = C \cdot \iota_2 \delta_l^{-2} \epsilon_l^{-2}$ for some absolute constant $C > 0$, $\epsilon_0 = \epsilon/H^2$ and $\epsilon_l = 2\epsilon_{l-1}$ for $l \geq 1$. $\delta_l = \max(\delta, \epsilon_l)$ is a threshold for *distinguishable* state-actions $x$ such that $\Delta(x) \geq \delta_l$. In a nut-shell, $\mathcal{E}_l$ is a set of sequences of state-actions in which every pair of state-actions has been explored at least $n_l$ times. Then we split a set of trajectories into disjoint sets $\mathcal{E}_0' = \mathcal{E}_0$ and $\mathcal{E}_l' = \mathcal{E}_{l-1}^c \cap \mathcal{E}_l$ and for $l \in [L + 1]$, *i.e.,* a set of trajectories with all pairs visited more than $n_l$ and at least one pair visited less than $n_{l-1}$, where we let $L = O(\log(H/\epsilon))$ be the largest integer such that $n_L \geq C \cdot \iota_2$. Let $n_{L+1} = 0$ and $\epsilon_{L+1} = 1$.

Recall that our goal is to control the $l_1$-statistical distance between distributions of trajectories for any history-dependent policies that resulted from true and empirical models. With above machinery, we can, instead, bound the $l_1$ statistical distance between all trajectories as the following:

$$\|(\mathbb{P}^\pi - \hat{\mathbb{P}}^\pi)(\tau)\|_1 = \sum_{l=0}^{L+1} \sum_{\tau: x_{1:H} \in \mathcal{E}_l'} |\mathbb{P}^\pi(\tau) - \hat{\mathbb{P}}^\pi(\tau)| \leq \sum_{l=0}^{L+1} \sup_{\pi \in \Pi} \mathbb{P}^\pi(x_{1:H} \in \mathcal{E}_l') \cdot O(H\epsilon_l), \tag{10}$$

where $\mathbb{P}^\pi(x_{1:H} \in \mathcal{E}_l')$, is the probability that the random trajectory $\tau$ observed when following a policy $\pi$ is contained within the set $\mathcal{E}_l'$. The following lemma is critical in bounding the above inner sum for each fixed $l$:

**Lemma 4.1 (Eventwise Total Variance Discrepancy)** *For any $l \in \{0, 1, ..., L+1\}$ and any history-dependent policy $\pi \in \Pi$, we have:*

$$\sum_{\tau: x_{1:H} \in \mathcal{E}_l'} |\mathbb{P}^\pi(\tau) - \hat{\mathbb{P}}^\pi(\tau)| \leq \sup_{\pi \in \Pi} \mathbb{P}^\pi(x_{1:H} \in \mathcal{E}_l') \cdot O(H\epsilon_l). \tag{11}$$

The second key connection is the relation between $\sup_{\pi \in \Pi} \mathbb{P}^\pi(\mathcal{E}_l')$ and the data collected in pure-exploration phase. The following lemma connects them:

**Lemma 4.2 (Event Probability Bound by Pure Exploration)** *With probability at least $1 - \eta$ Algorithm 2 terminates after at most $K$ episodes where*

$$K \geq C \cdot S^2 A(H + A)\epsilon_{pe}^{-2} \log(HSA/(\epsilon_{pe}\eta)), \tag{12}$$

*with some absolute constant $C > 0$. Furthermore, for every $l \in [L + 1]$ we have*

$$\sup_{\pi \in \Pi} \mathbb{P}^\pi(x_{1:H} \in \mathcal{E}'_l) \leq O\left(H\epsilon_{pe}\delta_l^{-1}\epsilon_l^{-1}\right). \tag{13}$$

Given the above two lemmas, we set $\epsilon_{pe} = \epsilon\delta_0/H^3 L$. We then apply Lemma 4.2 and (10) to bound a difference in expected value of true and empirical models:

$$|V_\mathcal{M}^\pi - V_{\hat{\mathcal{M}}}^\pi| \leq H \cdot \|(\mathbb{P}^\pi - \hat{\mathbb{P}}^\pi)(\tau)\|_1 \leq \tilde{O}\left(H^3\epsilon_{pe}\delta_0^{-1}\right) \leq O(\epsilon).$$

Finally, plugging $\epsilon_{pe} = \epsilon\delta_0/H^3 L$ into the required number of episodes (12) gives a sample-complexity bound presented in Theorem 3.2. Full proof of Lemma 4.2 is given in Appendix B.3.

## 4.1 Key Ideas for Lemma 4.1

The core of analysis by which we establish Lemma 4.1 goes as follows. We split the state-action pairs into two sets *distinguishable* and *indistinguishable* state-action pairs. Fix an $l \in \{0, 1, ...L + 1\}$ and consider then set $\mathcal{E}'_l$. We call a state-action pair $x \in \mathcal{S} \times \mathcal{A}$ $\delta$-distinguishable if $\Delta(x)$ is larger then some $\delta > 0$. Then, by the definition of $\Delta(x)$, it implies that $p_1(x)$ and $p_2(x)$ are sufficiently different. For a trajectory $\tau \in \mathcal{E}'_l$, we say that a state-action $x$ within $\tau$ is $\delta_l$-distinguishable if $\Delta(x) \geq \delta_l$.

States that are distinguishable have the following property. If pairwise correlations between three distinguishable state-actions are well estimated, then we can recover individual reward probability for each of the three state-actions as in the following lemma:

**Lemma 4.3** *Suppose that $(x_1, x_2), (x_2, x_3), (x_3, x_1) \in \mathcal{X}_l$ and $\Delta(x_i) \geq \delta_l$ for all $i \in \{1, 2, 3\}$. Let $i^* = \arg\max_{i \in [3]} \Delta(x_i)$, and $i_2^* = arg\max_{i \in [3], i \neq i^*} \Delta(x_i)$. Then we have*

$$\forall i \neq i^*, |\hat{\Delta}(x_i) - \Delta(x_i)| \leq \epsilon_l/2, \text{ and } \left|\hat{\Delta}(x_{i^*}) - \Delta(x_{i^*})\right| \leq \frac{\Delta_{i^*}}{\Delta_{i_2^*}}\epsilon_l/2.$$

If all reward parameters in a trajectory are relatively well-estimated, then we can bound the error in the estimated and true probability of the trajectory. Thus, if a trajectory $\tau \in \mathcal{E}'_l$ contains more than two distinguishable state-actions, we can rely on the closeness in reward parameters to bound $|\mathbb{P}^\pi(\tau) - \hat{\mathbb{P}}^\pi(\tau)|$. On the other hand, if there are no such three distinguishable state-actions in $\tau$, then we cannot guarantee the correctness of reward parameters in $\tau$. In such case, we take a different path to show the closeness in trajectory distributions. Full proof is given in Appendix A.

## 5 Discussions and Future Work

We developed the first efficient algorithm for RL in two reward-mixing MDPs without any assumptions on system dynamics. We conclude this work with discussions on our assumptions and several future directions.

**Non-uniform/Unknown Mixing Weights.** When the prior over contexts is not uniform, *i.e.,* $w \neq 1/2$, parameter recovery procedure (Algorithm 3) may be more complicated due to the sign-ambiguity issue in $p_-(x)$. When $w = 1/2$, the estimated model is identical whether we use $\hat{p}_-(x) \approx p_-(x)$ or $\hat{p}_-(x) \approx -p_-(x)$. However, the model is no longer identical when we have correct signs of $p_-(x)$ or not if $w \neq 1/2$. In Appendix E, we discuss the extension to non-uniform mixing weights in a great detail. Once we can solve with any known $w \in [0, 1]$, then one possible solution to handle unknown mixing weight is to sweep all possible mixing weight for $w$ from 0 to 1 with discretization level $O(\epsilon/H)$, recover an empirical model obtained by solving an LP assuming known $w$ for all discretized weights and pick the best policy by testing each policy on the environment.

**Structured Reward Distributions.** Our approach can be extended to structured reward distributions, *e.g.,* distributions over finite supports. For instance, if a reward can take $l$ different values in $\{q_1, q_2, ..., q_l\}$, we redefine $\mu(x_1, x_2) := \frac{1}{2}\sum_{m=1}^{2} p_m(x_1)p_m(x_2)^\top$ as a $l \times l$ moment-matrix with setting $p_m(x) = [R_m(r = q_1|x), R_m(r = q_2|x), ..., R_m(r = q_l|x)]^\top$. Then, we can extend an LP in Algorithm 3 to find $\hat{p}_m(x)$ to match $\mu(x_1, x_2)$ within confidence intervals. This approach would also pay extra $poly(l)$-factors both in sample and computational complexity. Then, once we can verify that the recovered difference in reward models is close, *i.e.,* $\|\hat{p}_-(x) - p_-(x)\|_1 \le O(\epsilon_l)$, then the remaining proof for Theorem 3.2 would follow similar to the Bernoulli reward case. We leave a more thorough investigation in this direction as future work.

**Extensions to More Than Two Reward Models** $M > 2$**.** As mentioned earlier, RM-MDP with $M$ reward models is an MDP analogy to learning a mixture of $M$ product distributions over reward distributions. If *every higher-order correlation of interest between multiple state-actions can be estimated*, we may apply existing methods for learning a mixture of product distributions over a binary field [20, 19, 26]. For instance, Jain and Oh [26] proposed a spectral method which requires all off-diagonal elements of up to third order moments to learn parameters of $M$-mixtures of $n$-product distributions. To apply their method, we may estimate correlations of all three different state-actions as similarly in Algorithm 2, then reward model recovery of RM-MDP can be converted to learning a mixture of $n = SA$-product reward distributions. However, as we have already seen in this paper, the real challenge is in recovering the model from correlations of *unevenly* reachable sets of state-actions, and then analyzing errors with different levels of uncertainty in correlations. We believe this is a very interesting direction for future research.

In addition to the above mentioned issues, we believe there are a number of possible future directions. While we specifically focused on RM-MDP problems, the more general problem of Latent MDP [36] has not been resolved even for $M = 2$ without any assumptions. We believe that it would be an interesting future direction to develop efficient algorithms for Latent MDPs with small $M$. We can also consider more technical questions in RM-MDPs, *e.g.,* what would be the optimal sample-complexity dependency on $H, \epsilon$, and whether we can develop a fully online algorithm to minimize regret. We leave them as future work.

### Acknowledgement

The research was funded by NSF grant 2019844, and by the Army Research Office and was accomplished under Cooperative Agreement Number W911NF-19-2-0333. The views and conclusions contained in this document are those of the authors and should not be interpreted as representing the official policies, either expressed or implied, of the Army Research Office or the U.S. Government. The U.S. Government is authorized to reproduce and distribute reprints for Government purposes notwithstanding any copyright notation herein.

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
