# Appendix A  Proof of Lemma 4.1

## A.1  Notation

We define new notation that will be utilized across the appendix. Let $\mathcal{E}$ be a subset of trajectories such that for some mapping $f : (\mathcal{S} \times \mathcal{A})^{\otimes H} \to \{0,1\}$, let $\mathcal{E} = \{x_{1:H}|f(x_{1:H}) = 1\}$. Let us define $\mathbb{P}^*_{\mathcal{E}}(h_t)$ be the maximum probability of getting any final trajectory $\tau = (x,r)_{1:H}$ that belongs to $\mathcal{E}$ starting from a history $h_t = ((x,r)_{1:t-1}, s_t)$. That is,

$$\mathbb{P}^*_{\mathcal{E}}(h_t) = \sup_{\pi \in \Pi} \mathbb{P}^\pi(x_{1:H} \in \mathcal{E}|h_t). \tag{14}$$

At the beginning of the episode, $h_0 = \phi$ and we denote $\mathbb{P}^*_{\mathcal{E}}(\phi)$ to mean $\sup_{\pi \in \Pi} \mathbb{P}^\pi(x_{1:H} \in \mathcal{E})$. We often use $\mathbb{P}^\pi(\mathcal{E})$ instead of $\mathbb{P}^\pi(x_{1:H} \in \mathcal{E})$, omitting $x_{1:H}$ when the context is clear.

We use $\pi_{1:t}$ for any $t \in [H]$ to denote $\Pi^t_{t'=1}\pi(a_{t'}|h_{t'})$. Similarly, with a slight abuse in notation, we define $T_{0:t} := \nu(s_1) \cdot \Pi^t_{t'=1}T(s_{t'+1}|s_{t'}, a_{t'})$ for $1 \le t \le H - 1$. When we define a new RM-MDP model $\mathcal{M}^c$ with a superscript $c \in \mathbb{N}$, we add a superscript $c$ for all quantities measured with respect to $\mathcal{M}^c$. For instance, reward probability at state $s$ and action $a$ in the $m^{th}$ in a model $\mathcal{M}^a$ is denoted as $R^c_m(\cdot|s,a)$, a probability of an event in a model $\mathcal{M}^a$ with a policy $\pi$ is denoted as $\mathbb{P}^{c,\pi}(\cdot)$.

In order to express the probability of a trajectory without conditioning on the context, we first introduce a convenient matrix form of probability $D(r|x) = \begin{bmatrix} R_1(r|x) & 0 \\ 0 & R_2(r|x) \end{bmatrix}$. Probability of observing a trajectory $\tau = (x,r)_{1:H}$ is then

$$\mathbb{P}^\pi(\tau) = \frac{1}{2}\pi_{1:t}T_{0:t} \cdot \mathbf{1}^\top \left(\Pi^H_{t=1}D(r_t|x_t)\right)\mathbf{1},$$

where $\mathbf{1}$ is the all-one vector $[11]^\top$. For any $t_1, t_2 \in [H]$, let $D_{t_1:t_2}$ be a short hand for $\Pi^{t_2}_{t=t_1}D(r_t|x_t)$. We use $(\cdot)_{t_1:t_2}$ in a similar manner for any other symbols. If $t = t_1 = t_2$, then we simply use $(\cdot)_t$. For instance, we use $D_t = D(r_t|x_t) = \begin{bmatrix} R_1(r_t|x_t) & 0 \\ 0 & R_2(r_t|x_t) \end{bmatrix}$.

## A.2  Proof of Lemma 4.1

We provide the outline of the proof in this section. All omitted proofs can be found in Appendix D. To simplify the presentation, we temporarily assume that we know the transition and initial state probabilities, *i.e.*, $\hat{T} = T, \hat{\nu} = \nu$. In Appendix B.4, we provide the full proof without assuming known transition and initial state probabilities, but intead using $\hat{\nu}$ and $\hat{T}$ estimated in Algorithm 2.

The analysis begins with the following lemma on the summation of target quantity when differences in two models are small parameter-wise.

**Lemma A.1** *Suppose that two 2RM-MDPs $\mathcal{M}^1, \mathcal{M}^2$ have the same transition kernel and initial distribution, and satisfy $\|(R^1_m - R^2_m)(r|x)\|_1 \le \epsilon_r$ for some $\epsilon_r > 0$ and any $m \in \{1,2\}$ and $x \in \mathcal{X}$ for a given set $\mathcal{X} \subseteq \mathcal{S} \times \mathcal{A}$. For any subset of length $H$ state-action sequences $\mathcal{E} \subseteq \mathcal{X}^{\otimes H}$, we have*

$$\sup_{\pi \in \Pi} \sum_{\tau:x_{1:H} \in \mathcal{E}} |\mathbb{P}^{1,\pi}(\tau) - \mathbb{P}^{2,\pi}(\tau)| \le \sup_{\pi \in \Pi} \mathbb{P}^{1,\pi}(\mathcal{E}) \cdot H\epsilon_r,$$

*where $\mathbb{P}^{c,\pi}(\cdot)$ is a probability measured in an environment modeled by $\mathcal{M}^c$ for $c = 1, 2$.*

Given Lemma A.1, we first consider surrogate 2RM-MDPs for the true and estimated models which ignore small differences in reward functions. Specifically, for all $x \in \mathcal{S} \times \mathcal{A}$, we define $\mathcal{M}^1$ that approximates the true model as:

$$\mathcal{M}^1 : p^1_m(x) = \epsilon_l + (1 - 2\epsilon_l)p_+(x), \quad m \in \{1,2\}, \qquad \text{if } \Delta(x) < 2\epsilon_l,$$
$$p^1_m(x) = \epsilon_l + (1 - 2\epsilon_l)p_m(x), \quad m \in \{1,2\}, \qquad \text{if } \Delta(x) \ge 2\epsilon_l, \tag{15}$$

and similarly $\mathcal{M}^2$ for the estimated model:

$$\mathcal{M}^2 : p^2_m(x) = \epsilon_l + (1 - 2\epsilon_l)p_+(x), \quad m \in \{1,2\}, \qquad \text{if } \hat{\Delta}(x) < 2\epsilon_l,$$

$$\begin{cases} p_1^2(x) = \epsilon_l + (1 - 2\epsilon_l)(p_+ + \hat{p}_-)(x) \\ p_2^2(x) = \epsilon_l + (1 - 2\epsilon_l)(p_+ - \hat{p}_-)(x) \end{cases}, \qquad \text{if } \hat{\Delta}(x) \geq 2\epsilon_l. \qquad (16)$$

Recall that reward models are determined by $R_m^c(r = 1|x) = p_m^c(x)$ and $R_m^c(r = 0|x) = 1 - p_m^c(x)$ for $m = 1, 2$ and $c = 1, 2$. In the above construction, note that for the average value we use the same quantity $p_+$ from the true model to ignore small differences in averaged rewards. In particular, note that $p_m^1(x) = p_m^2(x)$ for any $x$ with $\Delta(x) < 2\epsilon_l$ and $\hat{\Delta}(x) < 2\epsilon_l$. For both models, we use true transition and initial state distribution models $T$ and $\nu$. By construction $\mathcal{M}^1$ and $\mathcal{M}^2$ well approximate $\mathcal{M}^*$ and $\hat{\mathcal{M}}$ respectively:

**Lemma A.2** *For all $m \in \{1, 2\}$ and $\forall x : \exists (x_i, x_j) \in \mathcal{X}_l$ s.t. $x = x_i$ or $x_j$, we have*

$$\|(R_m - R_m^1)(r|x)\|_1 \leq 4\epsilon_l, \quad \|(\hat{R}_m - R_m^2)(r|x)\|_1 \leq 4\epsilon_l.$$

Then using Lemma A.1, we have

$$\sum_{\tau : x_{1:H} \in \mathcal{E}_l'} |\mathbb{P}^\pi(\tau) - \hat{\mathbb{P}}^\pi(\tau)| \leq \sup_{\pi \in \Pi} \mathbb{P}^\pi(\mathcal{E}_l') \cdot O(H\epsilon_l) + \sum_{\tau : x_{1:H} \in \mathcal{E}_l'} |\mathbb{P}^{1,\pi}(\tau) - \mathbb{P}^{2,\pi}(\tau)|,$$

where we recall that $\mathcal{E}_l'$ is defined as $\mathcal{E}_0' = \mathcal{E}_0$, and $\mathcal{E}_l' = \mathcal{E}_{l-1}^c \cap \mathcal{E}_l$ for $l \geq 1$.

Now we continue the discussion in Section 4.1. We divide a set of trajectories $\mathcal{E}_l'$ by whether the number of $\delta_l$-distinguishable pairs in a trajectory $\tau : x_{1:H} \in \mathcal{E}_l'$ is less than 3, *i.e.*, $\sum_{t=1}^H \mathbb{1}\{\Delta(x_t) \geq \delta_l\}$ being $\leq 2$ or $\geq 3$:

$$\mathcal{E}_{l,2} = \left\{ x_{1:H} \in \mathcal{E}_l' \Big| \sum_{t=1}^H \mathbb{1}\{\Delta(x_t) \geq \delta_l\} \leq 2 \right\},$$

$$\mathcal{E}_{l,3} = \left\{ x_{1:H} \in \mathcal{E}_l' \Big| \sum_{t=1}^H \mathbb{1}\{\Delta(x_t) \geq \delta_l\} \geq 3 \right\}. \qquad (17)$$

We handle each case separately and show that $\sum_{\tau : x_{1:H} \in \mathcal{E}} |\mathbb{P}^{1,\pi}(\tau) - \mathbb{P}^{2,\pi}(\tau)| \leq \sup_{\pi \in \Pi} \mathbb{P}^\pi(\mathcal{E}) \cdot O(H\epsilon_l)$ for $\mathcal{E} = \mathcal{E}_{l,2}$ and $\mathcal{E}_{l,3}$ respectively.

### A.2.1 Case I: $\mathcal{E}_{l,3}$.

For any $\tau : x_{1:H} \in \mathcal{E}_{l,3}$, let any $t_1, t_2, t_3 \in [H]$ such that $\Delta(x_{t_i}) \geq \delta_l$ for $i = 1, 2, 3$. From Lemma 4.3, we obtain a corollary on parameters of all state-actions that appear in $\tau : x_{1:H} \in \mathcal{E}_{l,3}$:

**Corollary 1** *For any $\tau : x_{1:H} \in \mathcal{E}_{l,3}$ and for every $t \in [H]$, let $t^* = \arg\max_{t \in [H]} \Delta(x_t)$, and $t_2^* = \arg\max_{t \in [H], t \neq t^*} \Delta(x_t)$. Then we have*

$$|\hat{\Delta}(x_t) - \Delta(x_t)| \leq 3\epsilon_l \quad \forall t \neq t^*, \qquad \left|\hat{\Delta}(x_{t^*}) - \Delta(x_{t^*})\right| \leq \frac{\Delta_{t^*}}{\Delta_{t_2^*}}\epsilon_l/2.$$

That is, in all trajectories in $\mathcal{E}_{l,3}$, all visited state-actions have good estimates of reward probabilities within $O(\epsilon_l)$-error (except at most one state-action $x_{t^*}$, which needs an extra care). In particular, we work with the fact that $|\Delta^1(x_t) - \Delta^2(x_t)| < O(\epsilon_l)$ for most of $t \in [H]$. This case is handled in Appendix A.4.

### A.2.2 Case II: $\mathcal{E}_{l,2}$.

In this case, the following lemma is the key result to bound the error in this case.

**Lemma A.3** *For any $x_{1:H} \in \mathcal{E}_{l,2}$, we have $\sum_{t=1}^H \mathbb{1}\left\{\Delta(x_t) \geq \epsilon_l \cup \hat{\Delta}(x_t) \geq 2\epsilon_l\right\} \leq 2$.*

Lemma A.3 ensures that for any $x_{1:H} \in \mathcal{E}_{l,2}$, except at most for some two time steps $t_1, t_2 \in [H]$, we have $\Delta(x_t), \hat{\Delta}(x_t)$ indistinguishable for all $t \neq t_1, t_2$. For such $t \neq t_1, t_2$, since an average of

rewards can be well-estimated, *i.e.*, $p_+(x_t) \approx \hat{p}_+(x_t)$, let us for now ignore the difference at $x_t$: $R_m(\cdot|x_t) \approx \hat{R}_m(\cdot|x_t)$ for $m = 1, 2$. Then for any $\tau = (x, r)_{1:H}$ for which $x_{1:H}$ belongs to $\mathcal{E}_{l,2}$:

$$|\mathbb{P}^\pi(\tau) - \hat{\mathbb{P}}^\pi(\tau)| \propto \mathbb{P}^\pi(\tau) \cdot |\mu(x_{t_1}, x_{t_2}) - \hat{\mu}(x_{t_1}, x_{t_2})| \leq O(\epsilon_l),$$

where $\mu(x_{t_1}, x_{t_2})$ is from equation (1), and the last inequality is due to the construction of $\hat{\mathcal{M}}$ which is designed to match in second-order correlations. Building upon the above idea, we show that $\sum_{\tau:x_{1:H} \in \mathcal{E}_{l,2}} |\mathbb{P}^\pi(\tau) - \hat{\mathbb{P}}^\pi(\tau)| \leq O(H\epsilon_l) \cdot \sup_{\pi \in \Pi} \mathbb{P}^\pi(\mathcal{E}_{l,2})$. This case is handled in Appendix A.5. Once we prove for $\mathcal{E}_{l,2}$ and $\mathcal{E}_{l,3}$ such that

$$\sum_{\tau:x_{1:H} \in \mathcal{E}_{l,3}} |\mathbb{P}^{1,\pi}(\tau) - \mathbb{P}^{2,\pi}(\tau)| \leq O(H\epsilon_l) \cdot \sup_{\pi \in \Pi} \mathbb{P}^\pi(\mathcal{E}_{l,3}), \tag{18}$$

$$\sum_{\tau:x_{1:H} \in \mathcal{E}_{l,2}} |\mathbb{P}^{1,\pi}(\tau) - \mathbb{P}^{2,\pi}(\tau)| \leq O(H\epsilon_l) \cdot \sup_{\pi \in \Pi} \mathbb{P}^\pi(\mathcal{E}_{l,2}), \tag{19}$$

then, since $\mathcal{E}_{l,2} \cup \mathcal{E}_{l,3} = \mathcal{E}'_l$, we obtain $\sum_{\tau:x_{1:H} \in \mathcal{E}'_l} |\mathbb{P}^\pi(\tau) - \hat{\mathbb{P}}^\pi(\tau)| \leq O(H\epsilon_l) \cdot \sup_{\pi \in \Pi} \mathbb{P}^\pi(\mathcal{E}'_l)$, which concludes Lemma 4.1.

## A.3 Proof of Lemma A.1

By the definition of $\mathbb{P}^*_\mathcal{E}(\cdot)$ defined in (14), we have the following inequalities: for any length $t$ history $h_t = ((s, a, r)_{1:t-1}, s_t)$, action $a_t$ and any history-dependent policy $\pi$, we have

$$\mathbb{P}^*_\mathcal{E}(h_t) \geq \sum_{a_t} \mathbb{P}^*_\mathcal{E}(h_t, a_t)\pi(a_t|h_t), \quad t < H, \tag{20}$$

$$\mathbb{P}^*_\mathcal{E}(h_t, a_t) \geq \sum_{s_{t+1}} \mathbb{P}^*_\mathcal{E}(h_{t+1})T(s_{t+1}|s_t, a_t), \quad t < H, \tag{21}$$

$$\mathbb{P}^*_\mathcal{E}(h_H) \geq \sum_{a_H} \mathbb{1}\{x_{1:H} \in \mathcal{E}\}\pi(a_H|h_H), \quad t = H. \tag{22}$$

Also, since $\mathbb{P}_{\mathcal{E}^*} = \sup_\pi \mathbb{P}^\pi(\tau \in \mathcal{E})$ only depends on the occurance of $x_{1:H}$, any two 2RM-MDP models with the same transition and initial distribution have the same value for $\mathbb{P}_{\mathcal{E}^*}$:

$$\mathbb{P}^*_\mathcal{E}(h) = \sup_{\pi \in \Pi} \mathbb{P}^{1,\pi}(x_{1:H} \in \mathcal{E}|h) = \sup_{\pi \in \Pi} \mathbb{P}^{2,\pi}(x_{1:H} \in \mathcal{E}|h).$$

Hence when we consider the same transition model, we often omit 1 and 2 in superscript from $\mathbb{P}^{1,\pi}(\mathcal{E}|h)$ or $\mathbb{P}^{2,\pi}(\mathcal{E}|h)$.

*Proof.* Now we prove Lemma A.1. Our target is to analyze the difference in the following:

$$\mathbb{P}^{c,\pi}(\tau) = \nu(s_1)\left(\Pi_{t=1}^H \pi(a_t|h_t)\right) \cdot \left(\Pi_{t=1}^{H-1} T(s_{t+1}|s_t, a_t)\right) \cdot \frac{1}{2}\mathbf{1}^\top D^c_{H:1}\mathbf{1},$$

for $c = 1, 2$ for all $\tau : x_{1:H} \in \mathcal{E}$.

We prove the lemma by backward induction. Let us denote $h_t = ((s, a, r)_{1:t-1}, s_t)$. Note that we assume here for any $x_t = (s_t, a_t)$ that appear in a trajectory $x_{1:H} \in \mathcal{E}$ at any time $t \in [H]$ satisfies $\|D^1(r_t|x_t) - D^2(r_t|x_t)\|_1 < \epsilon_r$. Hence,

$$2 \cdot \sum_{\tau:x_{1:H} \in \mathcal{E}} |\mathbb{P}^{1,\pi}(\tau) - \mathbb{P}^{2,\pi}(\tau)|$$

$$= \sum_{h_H} \sum_{a_H, r_H} \mathbb{1}\{x_{1:H} \in \mathcal{E}\}\pi(a_H|h_H)\pi_{1:H-1}T_{0:H-1}|\mathbf{1}^\top(D^1_{H:1} - D^2_{H:1})\mathbf{1}|$$

$$\leq \sum_{h_H} \sum_{a_H, r_H} \mathbb{1}\{x_{1:H} \in \mathcal{E}\}\pi(a_H|h_H)\pi_{1:H-1}T_{0:H-1}\|(D^1_H - D^2_H) \cdot D^1_{H-1:1}\mathbf{1}\|_1$$

$$+ \sum_{h_H} \sum_{a_H} \mathbb{1}\{x_{1:H} \in \mathcal{E}\}\pi(a_H|h_H)\pi_{1:H-1}T_{0:H-1}\|D^2_H(D^1_{H-1:1} - D^2_{H-1:1})\mathbf{1}\|_1$$

$$\leq \sum_{h_H} \sum_{a_H} \mathbb{1}\left\{x_{1:H} \in \mathcal{E}\right\} \pi(a_H|h_H)\pi_{1:H-1}T_{0:H-1}\|D^1_{H-1:1}\mathbf{1}\|_1\epsilon_r$$

$$+ \sum_{h_H} \sum_{a_H} \mathbb{1}\left\{x_{1:H} \in \mathcal{E}\right\} \pi(a_H|h_H)\pi_{1:H-1}T_{0:H-1}\|(D^1_{H-1:1} - D^2_{H-1:1})\mathbf{1}\|_1$$

$$\leq \epsilon_r \sum_{h_H} \mathbb{P}^*_{\mathcal{E}}(h_H)\mathbb{P}^{1,\pi}(h_H) + \sum_{h_H} \mathbb{P}^*_{\mathcal{E}}(h_H)\pi_{1:H-1}T_{0:H-1}\|(D^1_{H-1:1} - D^2_{H-1:1})\mathbf{1}\|_1.$$

The first term in the last inequality above can be bounded as the summation over all possible length $H$ histories such that for any policy $\pi \in \Pi$, we have

$$\mathbb{P}^*_{\mathcal{E}}(\phi) = \sup_{\pi} \mathbb{P}^{\pi}(\mathcal{E}) = \sup_{\pi} \sum_{h_H} \mathbb{P}^{\pi}(\mathcal{E}, h_H) = \sup_{\pi} \sum_{h_H} \mathbb{P}^{\pi}(\mathcal{E}|h_H)\mathbb{P}^{\pi}(h_H) = \sup_{\pi} \sum_{h_H} \mathbb{P}^*_{\mathcal{E}}(h_H)\mathbb{P}^{\pi}(h_H).$$

Then we can proceed from time step $t = H$ to $t = H - 1$:

$$\sum_{h_H} \mathbb{P}^*_{\mathcal{E}}(h_H)\pi_{1:H-1}T_{0:H-1}\|(D^1_{H-1:1} - D^2_{H-1:1})\mathbf{1}\|_1$$

$$\leq \sum_{h_{H-1}} \sum_{a_{H-1}, r_{H-1}, s_H} \mathbb{P}^*_{\mathcal{E}}(h_H)\pi(a_{H-1}|h_{H-1})T(s_H|x_{H-1})\pi_{1:H-2}T_{0:H-2}\|(D^1_{H-1} - D^2_{H-1}) \cdot D^1_{H-2:1}\mathbf{1}\|_1$$

$$+ \sum_{h_{H-1}} \sum_{a_{H-1}, s_H} \mathbb{P}^*_{\mathcal{E}}(h_H)\pi(a_{H-1}|h_{H-1})T(s_H|x_{H-1})\pi_{1:H-2}T_{0:H-2}\|(D^1_{H-2:1} - D^2_{H-2:1})\mathbf{1}\|_1$$

$$\leq \epsilon_r \cdot \sum_{h_{H-1}} \mathbb{P}^{1,\pi}(h_{H-1}) \sum_{a_{H-1}, s_H} \mathbb{P}^*_{\mathcal{E}}(h_H)\pi(a_{H-1}|h_{H-1})T(s_H|x_{H-1})$$

$$+ \sum_{h_{H-1}} \pi_{1:H-2}T_{0:H-2}\|(D^1_{H-2:1} - D^2_{H-2:1})\mathbf{1}\|_1 \sum_{a_{H-1}, s_H} \mathbb{P}^*_{\mathcal{E}}(h_H)\pi(a_{H-1}|h_{H-1})T(s_H|x_{H-1})$$

$$\leq \mathbb{P}^*_{\mathcal{E}}(\phi)\epsilon_r + \sum_{h_{H-1}} \mathbb{P}^*_{\mathcal{E}}(h_{H-1})\pi_{1:H-2}T_{0:H-2}\|(D^1_{H-2:1} - D^2_{H-2:1})\mathbf{1}\|_1,$$

where the last inequality comes from (22). We can repeat this procedure until we reach $t = 1$ in backwards. Note that $\mathbb{P}^*_{\mathcal{E}}(\phi) = \sup_{\pi} \mathbb{P}^{\pi}(\mathcal{E})$, which gives Lemma A.1. $\qquad \square$

### A.4 Analysis for Case I: $\mathcal{E}_{l,3}$ (equation (18))

#### A.4.1 Equivalence in Signs

Before we start the proof, we need to point out one important fact. For any $x_{1:H} \in \mathcal{E}_{l,3}$, for all $x \in \{x_t\}_{t=1}^H$ such that $\Delta(x) \geq 2\epsilon_l$, either one of the two is true: $sign(p_-(x)) = sign(\hat{p}_-(x))$ or $sign(p_-(x)) = -sign(\hat{p}_-(x))$. This then implies $sign(p^1_-(x)) = sign(p^2_-(x))$ or $sign(p^1_-(x)) = -sign(p^2_-(x))$ whenever $|p^1_-(x)| > 0$ and $|p^2_-(x)| > 0$ consistently for all state-actions $x$ of interest. This can be shown by a simple contradiction argument: suppose there exists $t_1, t_2 \in [H]$ such that $t_1 \neq t_2$ and $\Delta(x_{t_1}), \Delta(x_{t_2}) \geq 2\epsilon_l$. If $sign(\hat{p}_-(x_{t_1})\hat{p}_-(x_{t_2})) \neq sign(p_-(x_{t_1})p_-(x_{t_2}))$, then this implies

$$|p_-(x_{t_1})p_-(x_{t_2}) - \hat{p}_-(x_{t_1})\hat{p}_-(x_{t_2})| \geq \max(\delta^2, 4\epsilon_l^2) > \epsilon_l\delta_l.$$

This violates the confidence interval constraint (7) since $n(x_{t_1}, x_{t_2}) \geq n_l \geq C \cdot \iota_2 \delta_l^{-2}\epsilon_l^{-2}$ and thus $b(x_{t_1}, x_{t_2}) < 0.01\delta_l\epsilon_l$, which forces $|p_-(x_{t_1})p_-(x_{t_2}) - \hat{p}_-(x_{t_1})\hat{p}_-(x_{t_2})| < \delta_l\epsilon_l$.

Now if this is the case, then without loss of generality, we can assume $sign(p^1_-(x)) = sign(p^2_-(x))$ for $x : |p^1_-(x)|, |p^2_-(x)| > 0$ since

$$\mathbb{P}^{2,\pi}(\tau) = \frac{1}{2}\pi_{1:H}T_{0:H-1}\mathbf{1}^\top D^2_{H:1}\mathbf{1}$$

$$= \frac{1}{2}\pi_{1:H}T_{0:H-1}\left(\Pi_{t=1}^H(p^2_+(x_t) + p^2_-(x_t)) + \Pi_{t=1}^H(p^2_+(x_t) - p^2_-(x_t))\right),$$

remains the same after we replace $p^2_-(x)$ by $-p^2_-(x)$ for all $x \in \{x_t\}_{t=1}^H$. This means, regardless of the sign of $p^2_-(x)$, the probability of any trajectories, which we eventually need, remains the

same. Hence now, without loss of generality, we assume that $sign(p^1_-(x)) = sign(p^2_-(x))$ for all $x \in \{x_t\}^H_{t=1}$ for any $x_{1:H} \in \mathcal{E}_{l,3}$. (Note that for non-uniform mixing weights, we need an extra care since $\mathbb{P}^{2,\pi}(\tau)$ is no more symmetric in signs of $p^2_-(x)$. See Appendix E to see the discussion on handling non-uniform priors).

Once the above holds, we can claim that

$$\|D^1(r=1|x) - D^2(r=1|x)\|_1 \leq O(\epsilon_l),$$

where $\|\cdot\|_1$ is a matrix $l_1$ norm here, for all $x \in \{x_t\}^H_{t=1}$. To see this, first note that

$$|R^1_1(r=1|x) - R^2_1(r=1|x)| = |p^1_1(x) - p^2_1(x)|$$
$$= |(p^1_+(x) + p^1_-(x)) - (p^2_+(x) + p^2_-(x))| = |p^1_-(x) - p^2_-(x)|,$$

where the last equality comes from the construction (15), (16) such that $p^1_+(x) = p^2_+(x)$. Since the sign of $p^1_-(x)$ is equal to the sign of $p^2_-(x)$, we have

$$|p^1_-(x) - p^2_-(x)| = |\Delta^1(x) - \Delta^2(x)|.$$

The same argument holds for $r = 0$. Note that (again from the construction), $|\Delta^1(x) - \Delta^2(x)| \leq |\Delta(x) - \hat{\Delta}(x)| + 4\epsilon_l$ and thus whenever $\hat{\Delta}(x)$ is close to $\Delta(x)$, we have small error in $D^1(r|x)$ and $D^2(r|x)$ as well.

### A.4.2 Notation

Before we proceed, we define a few more notation here. We occasionally use $R_+(r|x) = \frac{1}{2}(R^1_1 + R^1_2)(r|x)$, and $R^c_-(r|x) = \frac{1}{2}(R^c_1 - R^c_2)(r|x)$ for $c = 1, 2$. Let $D_+(r|x)$ and $D_-(r|x)$ for all $r \in \{0, 1\}$ and $x \in \mathcal{S} \times \mathcal{A}$ as

$$D_+(r|x) := \frac{1}{2}(D^1 + D^2)(r|x) = R_+(r|x) \begin{bmatrix} 1 & 0 \\ 0 & 1 \end{bmatrix}, \tag{23}$$

$$D_-(r|x) := \frac{1}{2}(D^1 - D^2)(r|x) = \frac{1}{2}(R^1_- - R^2_-)(r|x) \begin{bmatrix} 1 & 0 \\ 0 & -1 \end{bmatrix}, \tag{24}$$

We also let $t^*, t^*_2$ as defined in Corollary 1:

$$t^* = \arg\max_{t \in [H]} \Delta(x_t), \qquad t^*_2 = \arg\max_{t \in [H]/\{t^*\}} \Delta(x_t).$$

Let us consider a set of state-actions $\mathcal{X}_{l,3}$ such that for all $\tau \in \mathcal{E}_{l,3}$, they are always the only maximum distinguishable state-actions whenever they are included in a trajectory. Formally, we consider

$$\mathcal{X}_{l,3} = \{x | \forall x_{1:H} \in \mathcal{E}_{l,3} : x \notin \{x_t\}^H_{t=1} \text{ or } \exists t^* \in [H] \text{ s.t. } x_{t^*} = x \text{ and } \Delta(x_t) \leq \Delta(x)/2, \forall t \neq t^*\}. \tag{25}$$

Note that for all other state-actions that are not in $\mathcal{X}_{l,3}$ and appear in any trajectories in $\mathcal{E}_{l,3}$, the error between true $\Delta$ and estimated $\hat{\Delta}$ is less than $2\epsilon_l$ by Corollary 1. We define a set of trajectories

$$\mathcal{E}_{l,4} = \{x_{1:H} \in \mathcal{E}_{l,3} | \exists t^* \in [H], x_{t^*} \in \mathcal{X}_{l,3}\}, \tag{26}$$

that contains one of state-actions in $\mathcal{X}_{l,3}$. For any $x_{1:H} \in \mathcal{E}_{l,3}$, we replace $D^1(\cdot|x_{t^*})$ with $D_+(\cdot|x_{t^*}) + D_-(\cdot|x_{t^*})$ and $D^2(\cdot|x_{t^*})$ with $D_+(\cdot|x_{t^*}) - D_-(\cdot|x_{t^*})$.

### A.4.3 Main Proof

Now we focus on bounding the sum of errors in predictions of trajectories in $\mathcal{E}_{l,3}$. Our strategy is first to split trajectories into $\mathcal{E}_{l,4}$ and $\mathcal{E}_{l,3} \cap \mathcal{E}^c_{l,4}$:

$$2 \cdot \sum_{\tau \in \mathcal{E}_{l,3}} |\mathbb{P}^{1,\pi}(\tau) - \mathbb{P}^{2,\pi}(\tau)| = \sum_{(x,r)_{1:H}} \mathbb{1}\{x_{1:H} \in \mathcal{E}_{l,3}\} T_{0:H-1} \pi_{1:H} |\mathbf{1}^\top (D^1_{H:1} - D^2_{H:1})\mathbf{1}|$$

$$= \sum_{(x,r)_{1:H}} \mathbb{1}\{x_{1:H} \in \mathcal{E}_{l,3} \cap \mathcal{E}^c_{l,4}\} T_{0:H-1} \pi_{1:H} |\mathbf{1}^\top (D^1_{H:1} - D^2_{H:1})\mathbf{1}|$$

$$+ \sum_{(x,r)_{1:H}} \mathbb{1}\{x_{1:H} \in \mathcal{E}_{l,4}\} T_{0:H-1}\pi_{1:H}|\mathbf{1}^\top(D^1_{H:1} - D^2_{H:1})\mathbf{1}|.$$

For the first term, by Corollary 1 for all $r \in \{0,1\}$ and $x \notin \mathcal{X}_{l,3}$ we have

$$\|D^1(r|x) - D^2(r|x)\|_1 = \max_{m \in \{1,2\}} |R^1_m(r|x) - R^2_m(r|x)|$$

$$= |\Delta^1(x) - \Delta^2(x)| \le |\Delta(x) - \hat{\Delta}(x)| + 4\epsilon_l \le 8\epsilon_l.$$

Now, since all trajectories in $\mathcal{E}_{l,3} \cap \mathcal{E}^c_{l,4}$ does not contain any state-actions in $\mathcal{X}_{l,3}$, we can use Lemma A.1 to show that the first term is less than $\mathbb{P}^*_{\mathcal{E}_{l,3} \cap \mathcal{E}^c_{l,4}}(\phi) \cdot O(H\epsilon_l)$. Therefore we focus on bounding the second term, the sum of errors in predictions of trajectories in $\mathcal{E}_{l,4}$.

We introduce the following five new quantities:

$$D^{1,0}(\cdot|x) = \begin{cases} D_+(\cdot|x), & \text{if } x \in \mathcal{X}_{l,3} \\ D^1(\cdot|x), & \text{otherwise} \end{cases} \quad , \quad D^{2,0}(\cdot|x) = \begin{cases} D_+(\cdot|x), & \text{if } x \in \mathcal{X}_{l,3} \\ D^2(\cdot|x), & \text{otherwise} \end{cases} \quad ,$$

$$D^{1,1}(\cdot|x) = \begin{cases} D_-(\cdot|x), & \text{if } x \in \mathcal{X}_{l,3} \\ D^1(\cdot|x), & \text{otherwise} \end{cases} \quad , \quad D^{2,1}(\cdot|x) = \begin{cases} D_-(\cdot|x), & \text{if } x \in \mathcal{X}_{l,3} \\ D^2(\cdot|x), & \text{otherwise} \end{cases} \quad ,$$

$$D^3(\cdot|x) = \begin{cases} D_-(\cdot|x), & \text{if } x \in \mathcal{X}_{l,3} \\ D_+(\cdot|x), & \text{otherwise} \end{cases} \quad .$$

Then we can decompose the target quantity into three:

$$\sum_{x_{1:H} \in \mathcal{E}_{l,4}, r_{1:H}} T_{0:H-1}\pi_{1:H}|\mathbf{1}^\top(D^1_{H:1} - D^2_{H:1})\mathbf{1}| \le \sum_{x_{1:H} \in \mathcal{E}_{l,4}, r_{1:H}} T_{0:H-1}\pi_{1:H}|\mathbf{1}^\top(D^{1,0}_{H:1} - D^{2,0}_{H:1})\mathbf{1}|$$

$$+ \sum_{x_{1:H} \in \mathcal{E}_{l,4}, r_{1:H}} T_{0:H-1}\pi_{1:H}|\mathbf{1}^\top(D^{1,1}_{H:1} - D^3_{H:1})\mathbf{1}|$$

$$+ \sum_{x_{1:H} \in \mathcal{E}_{l,4}, r_{1:H}} T_{0:H-1}\pi_{1:H}|\mathbf{1}^\top(D^{2,1}_{H:1} - D^3_{H:1})\mathbf{1}|.$$

$$(27)$$

Since $\|D^{1,0}_t - D^{2,0}_t\|_1 \le O(\epsilon_l)$ for all $t$, the first term can be bounded by $\sup_\pi \mathbb{P}^*_{\mathcal{E}_{l,4}} \cdot O(H\epsilon_l)$ similarly to the case in Section A.3. For the second term, we proceed as the following:

$$\sum_{x_{1:H} \in \mathcal{E}_{l,4}, r_{1:H}} T_{0:H-1}\pi_{1:H}|\mathbf{1}^\top(D^{1,1}_{H:1} - D^3_{H:1})\mathbf{1}|$$

$$\le \underbrace{\sum_{h_H} \sum_{a_H, r_H} \mathbb{1}\{x_{1:H} \in \mathcal{E}_{l,4}\} \pi(a_H|h_H)\pi_{1:H-1}T_{0:H-1}\|(D^{1,1}_H - D^3_H) \cdot D^3_{H-1:1}\mathbf{1}\|_1}_{(i)}$$

$$+ \underbrace{\sum_{h_H} \sum_{a_H, r_H} \mathbb{1}\{x_{1:H} \in \mathcal{E}_{l,4}\} \pi(a_H|h_H)\pi_{1:H-1}T_{0:H-1}\|D^{1,1}_H \cdot (D^{1,1}_{H-1:1} - D^3_{H-1:1})\mathbf{1}\|_1}_{(ii)}.$$

We first aim to handle the term $(i)$. Here we can divide the cases such that when $x_H \notin \mathcal{X}_{l,3}$ and when $x_H \in \mathcal{X}_{l,3}$. In the former case, we first note that for any $x \notin \mathcal{X}_{l,3}$, we have

$$\|D^1(r|x) - D_+(r|x)\|_1 = \frac{1}{2}|R^1_1(r|x) - R^1_2(r|x)| = \Delta^1(x) \le \Delta(x),$$

where the last inequality is followed by the construction of $\mathcal{M}^1$. From the above, we have

$$\sum_{r_H} \mathbb{1}\{x_{1:H} \in \mathcal{E}_{l,4}\} \pi(a_H|h_H)\|(D^{1,1}_H - D^3_H)D^3_{H-1:1}\mathbf{1}\|_1 \le \Delta(x_H)\mathbb{1}\{x_{1:H} \in \mathcal{E}_{l,4}\} \pi(a_H|h_H)\|D^3_{H-1:1}\mathbf{1}\|_1.$$

In the latter case $x_H \in \mathcal{X}_{l,3}$, we have $D^{1,1}_H - D^3_H = 0$ by construction. Therefore (i) can be bounded as

$$(i) \le \sum_{h_H} \pi_{1:H-1}T_{0:H-1}\|D^3_{H-1:1}\mathbf{1}\|_1 \sum_{a_H : x_H \notin \mathcal{X}_{l,3}} \mathbb{1}\{x_{1:H} \in \mathcal{E}_{l,4}\} \Delta(x_H)\pi(a_H|h_H)$$

$$\leq 2 \sum_{h_H:\{x_t\}_{t=1}^{H-1}\cap\mathcal{X}_{l,3}\neq\emptyset} \pi_{1:H-1}T_{0:H-1}\mathbb{P}^*_{\mathcal{E}_{l,4}}(h_H)\epsilon_l\|(D_+)_{H-1:1}\mathbf{1}\|_1 \leq 2\epsilon_l\mathbb{P}^*_{\mathcal{E}_{l,4}}(\phi). \qquad (28)$$

The last inequality follows from the fact that since we consider trajectories in $\mathcal{E}_{l,4}$, if $x_H \notin \mathcal{X}_{l,3}$, then there is a unique $t^* \leq H-1$ such that $D_{t^*}^3 = c_{t^*}\begin{bmatrix} 1 & 0 \\ 0 & -1 \end{bmatrix}$ for $c_{t^*}$ that satisfies

$$c_{t^*} \leq \frac{1}{2}|\Delta^1(x_{t^*}) - \Delta^2(x_{t^*})| \leq \frac{1}{2}|\Delta(x_{t^*}) - \hat{\Delta}(x_{t^*})| + 2\epsilon_l \leq 2\epsilon_l \cdot \frac{\Delta(x_{t^*})}{\Delta(x_H)} \leq \frac{2\epsilon_l}{\Delta(x_H)},$$

where the last inequality follows from Corollary 1. Furthermore, for all other $t \neq t^* \leq H-1$ by construction we have $D_t^3 = (R_+)_t\begin{bmatrix} 1 & 0 \\ 0 & 1 \end{bmatrix}$. Therefore we have $\|D_{H-1:1}^3\mathbf{1}\|_1 = c_{t^*}\|(D_+)_{H-1:1}\mathbf{1}\|_1$. We plug this into (28) and apply the bound on $c_{t^*}$.

For the second term $(ii)$, we split into two cases when $x_H \notin \mathcal{X}_{l,3}$ and $x_H \in \mathcal{X}_{l,3}$, or equivalently, $\{x_t\}_{t=1}^{H-1}\cap\mathcal{X}_{l,3} \neq \emptyset$ and $\{x_t\}_{t=1}^{H-1}\cap\mathcal{X}_{l,3} = \emptyset$. In the former case, we simply reduce $(ii)$ such that

$$\sum_{r_H} \mathbb{1}\{x_{1:H}\in\mathcal{E}_{l,4}\}\|D_H^{1,1}(D_{H-1:1}^{1,1} - D_{H-1:1}^3)\mathbf{1}\|_1 \leq \mathbb{1}\{x_{1:H}\in\mathcal{E}_{l,4}\}\|(D_{H-1:1}^{1,1} - D_{H-1:1}^3)\mathbf{1}\|_1.$$

In order to handle the case $x_H \in \mathcal{X}_{l,3}$, let us define $m_t := \max_{t'<t}\Delta(x_{t'})$ for $t \geq 2$ and $m_1 = 1$. Since $D_H^{1,1} = (D_-)_H = c_H\begin{bmatrix} 1 & 0 \\ 0 & -1 \end{bmatrix}$ for some $c_H \leq 2\epsilon_l/m_H$, we have

$$\sum_{r_H} \mathbb{1}\{x_{1:H}\in\mathcal{E}_{l,4}\}\|D_H^{1,1}(D_{H-1:1}^{1,1} - D_{H-1:1}^3)\mathbf{1}\|_1 \leq 2\epsilon_l \cdot \frac{\mathbb{1}\{x_{1:H}\in\mathcal{E}_{l,4}\}}{m_H}\|(D_{H-1:1}^{1,1} - D_{H-1:1}^3)\mathbf{1}\|_1.$$

From the fact above, we can reduce $(ii)$ as the following:

$$(ii) \leq 2\epsilon_l \sum_{h_H:\{x_t\}_{t=1}^{H-1}\cap\mathcal{X}_{l,3}=\emptyset} \pi_{1:H-1}T_{0:H-1}\frac{\mathbb{P}^*_{\mathcal{E}_{l,4}}(h_H)}{m_H}\|(D_{H-1:1}^{1,1} - D_{H-1:1}^3)\mathbf{1}\|_1$$
$$+ \sum_{h_H:\{x_t\}_{t=1}^{H-1}\cap\mathcal{X}_{l,3}\neq\emptyset} \pi_{1:H-1}T_{0:H-1}\mathbb{P}^*_{\mathcal{E}_{l,4}}(h_H)\|(D_{H-1:1}^{1,1} - D_{H-1:1}^3)\mathbf{1}\|_1.$$

In order to proceed to the next time step, we need the following recursive relation for any $2 \leq t \leq H$:

$$2\epsilon_l \sum_{h_t:\{x_{t'}\}_{t'=1}^{t-1}\cap\mathcal{X}_{l,3}=\emptyset} \pi_{1:t-1}T_{1:t-1}\frac{\mathbb{P}^*_{\mathcal{E}_{l,4}}(x_{1:t-1},s_t)}{m_t}\|(D_{t-1:1}^{1,1} - D_{t-1:1}^3)\mathbf{1}\|_1$$
$$+ \sum_{h_t:\{x_{t'}\}_{t'=1}^{t-1}\cap\mathcal{X}_{l,3}\neq\emptyset} \pi_{1:t-1}T_{1:t-1}\mathbb{P}^*_{\mathcal{E}_{l,4}}(x_{1:t-1},s_t)\|(D_{t-1:1}^{1,1} - D_{t-1:1}^3)\mathbf{1}\|_1$$
$$\leq 4\epsilon_l\mathbb{P}^*_{\mathcal{E}_{l,4}}(\phi) + 2\epsilon_l \sum_{h_{t-1}:\{x_{t'}\}_{t=1}^{t-2}\cap\mathcal{X}_{l,3}=\emptyset} \pi_{1:t-2}T_{1:t-2}\frac{\mathbb{P}^*_{\mathcal{E}_{l,4}}(x_{1:t-2},s_{t-1})}{m_{t-1}}\|(D_{t-2:1}^{1,1} - D_{t-2:1}^3)\mathbf{1}\|_1$$
$$+ \sum_{h_{t-1}:\{x_{t'}\}_{t=1}^{t-2}\cap\mathcal{X}_{l,3}\neq\emptyset} \pi_{1:t-2}T_{1:t-2}\mathbb{P}^*_{\mathcal{E}_{l,4}}(x_{1:t-2},s_{t-1})\|(D_{t-2:1}^{1,1} - D_{t-2:1}^3)\mathbf{1}\|_1. \qquad (29)$$

By applying (29) recursively until $t = 2$, we conclude that $(ii) \leq 4H\epsilon_l \cdot \mathbb{P}^*_{\mathcal{E}_{l,4}}(\phi)$. Plugging this back to (27) (with a similar bound for the final term), we have

$$\sum_{x_{1:H}\in\mathcal{E}_{l,4},r_{1:H}} \nu(s_1)T_{1:H-1}\pi_{1:H}|\mathbf{1}^\top(D_{H:1}^1 - D_{H:1}^2)\mathbf{1}| \leq O(H\epsilon_l)\mathbb{P}^*_{\mathcal{E}_{l,4}}(\phi).$$

We conclude that $\sum_{\tau:x_{1:H}\in\mathcal{E}_{l,3}}|\mathbb{P}^{1,\pi}(\tau) - \mathbb{P}^{2,\pi}(\tau)| \leq O(H\epsilon_l) \cdot \mathbb{P}^*_{\mathcal{E}_{l,3}}(\phi)$.

### A.4.4 Proof of Equation (29)

We start with observing that for $(h_{t-1}, a_{t-1}) : \{x_{t'}\}_{t'=1}^{t-1} \cap \mathcal{X}_{l,3} = \emptyset$, where we can reduce the sum at time $t$ as

$$\sum_{r_{t-1}, s_t} \pi(a_{t-1}|h_{t-1})T(s_t|s_{t-1}, a_{t-1}) \frac{\mathbb{P}^*_{\mathcal{E}_{l,4}}(x_{1:t-1}, s_t)}{m_t} \|(D^{1,1}_{t-1:1} - D^3_{t-1:1})\mathbf{1}\|_1$$

$$\leq \sum_{r_{t-1}, s_t} \pi(a_{t-1}|h_{t-1})T(s_t|s_{t-1}, a_{t-1}) \frac{\mathbb{P}^*_{\mathcal{E}_{l,4}}(x_{1:t-1}, s_t)}{m_t} \|D^{1,1}_{t-1} \cdot (D^{1,1}_{t-2:1} - D^3_{t-2:1})\mathbf{1}\|_1$$

$$+ \sum_{r_{t-1}, s_t} \pi(a_{t-1}|h_{t-1})T(s_t|s_{t-1}, a_{t-1}) \frac{\mathbb{P}^*_{\mathcal{E}_{l,4}}(x_{1:t-1}, s_t)}{m_t} \|(D^{1,1}_{t-1} - D^3_{t-1}) \cdot D^3_{t-2:1}\mathbf{1}\|_1$$

$$\leq \sum_{s_t} \pi(a_{t-1}|h_{t-1})T(s_t|s_{t-1}, a_{t-1}) \frac{\mathbb{P}^*_{\mathcal{E}_{l,4}}(x_{1:t-1}, s_t)}{m_{t-1}} \|(D^{1,1}_{t-2:1} - D^3_{t-2:1})\mathbf{1}\|_1$$

$$+ \sum_{s_t} \pi(a_{t-1}|h_{t-1})T(s_t|s_{t-1}, a_{t-1})\mathbb{P}^*_{\mathcal{E}_{l,4}}(x_{1:t-1}, s_t) \|D^3_{t-2:1}\mathbf{1}\|_1.$$

where in the last inequality we used $m_t \geq m_{t-1}$ by definition. Note that $D^3_{t-2:1} = (D_+)_{t-2:1}$ by the construction of $D^3$.

For the case $(h_{t-1}, a_{t-1}) : \{x_{t'}\}_{t'=1}^{t-1} \cap \mathcal{X}_{l,3} \neq \emptyset$, we consider two cases $x_{t-1} \in \mathcal{X}_{l,3}$ and $x_{t-1} \notin \mathcal{X}_{l,3}$. In the former, we have the remaining terms for $h_{t-1} : \{x_{t'}\}_{t'=1}^{t-2} \cap \mathcal{X}_{l,3} = \emptyset$. That is,

$$\sum_{r_{t-1}, s_t} \pi(a_{t-1}|h_{t-1})T(s_t|s_{t-1}, a_{t-1})\mathbb{P}^*_{\mathcal{E}_{l,4}}(x_{1:t-1}, s_t) \|(D^{1,1}_{t-1:1} - D^3_{t-1:1})\mathbf{1}\|_1$$

$$\leq \sum_{r_{t-1}, s_t} \pi(a_{t-1}|h_{t-1})T(s_t|s_{t-1}, a_{t-1})\mathbb{P}^*_{\mathcal{E}_{l,4}}(x_{1:t-1}, s_t) \|D^{1,1}_{t-1} \cdot (D^{1,1}_{t-2:1} - D^3_{t-2:1})\mathbf{1}\|_1$$

$$\leq 2\epsilon_l \sum_{s_t} \pi(a_{t-1}|h_{t-1})T(s_t|s_{t-1}, a_{t-1}) \frac{\mathbb{P}^*_{\mathcal{E}_{l,4}}(x_{1:t-1}, s_t)}{m_{t-1}} \|(D^{1,1}_{t-2:1} - D^3_{t-2:1})\mathbf{1}\|_1,$$

where the last inequality can be derived as we did in (28).

In the latter case when $x_{t-1} \notin \mathcal{X}_{l,3}$, we first note that $\{x_{t'}\}_{t'=1}^{t-2} \cap \mathcal{X}_{l,3} \neq \emptyset$. In this case we proceed as the following:

$$\sum_{r_{t-1}, s_t} \pi(a_{t-1}|h_{t-1})T(s_t|s_{t-1}, a_{t-1})\mathbb{P}^*_{\mathcal{E}_{l,4}}(x_{1:t-1}, s_t) \|(D^{1,1}_{t-1:1} - D^3_{t-1:1})\mathbf{1}\|_1$$

$$\leq \sum_{r_{t-1}, s_t} \pi(a_{t-1}|h_{t-1})T(s_t|s_{t-1}, a_{t-1})\mathbb{P}^*_{\mathcal{E}_{l,4}}(x_{1:t-1}, s_t) \|D^{1,1}_{t-1} \cdot (D^{1,1}_{t-2:1} - D^3_{t-2:1})\mathbf{1}\|_1$$

$$+ \sum_{r_{t-1}, s_t} \pi(a_{t-1}|h_{t-1})T(s_t|s_{t-1}, a_{t-1})\mathbb{P}^*_{\mathcal{E}_{l,4}}(x_{1:t-1}, s_t) \|(D^{1,1}_{t-1} - D^3_{t-1}) \cdot D^3_{t-2:1}\mathbf{1}\|_1$$

$$\leq \sum_{s_t} \pi(a_{t-1}|h_{t-1})T(s_t|s_{t-1}, a_{t-1})\mathbb{P}^*_{\mathcal{E}_{l,4}}(x_{1:t-1}, s_t) \|(D^{1,1}_{t-2:1} - D^3_{t-2:1})\mathbf{1}\|_1$$

$$+ \sum_{s_t} \pi(a_{t-1}|h_{t-1})T(s_t|s_{t-1}, a_{t-1})\mathbb{P}^*_{\mathcal{E}_{l,4}}(x_{1:t-1}, s_t)\Delta(x_{t-1}) \|D^3_{t-2:1}\mathbf{1}\|_1.$$

Using the similar trick as in (28), we can reduce $\Delta(x_{t-1})\|D^3_{t-2:1}\mathbf{1}\|_1 \leq 2\epsilon_l \|(D_+)_{t-2:1}\mathbf{1}\|_1$ since in this trajectory there must exist $t^* \leq t - 2$ such that $x_{t^*} \in \mathcal{X}_{l,3}$. Now we need to combining all terms and sum over $h_{t-1}$:

$$2\epsilon_l \sum_{h_t : \{x_{t'}\}_{t'=1}^{t-1} \cap \mathcal{X}_{l,3} = \emptyset} \pi_{1:t-1}T_{1:t-1} \frac{\mathbb{P}^*_{\mathcal{E}_{l,4}}(x_{1:t-1}, s_t)}{m_t} \|(D^{1,1}_{t-1:1} - D^3_{t-1:1})\mathbf{1}\|_1$$

$$+ \sum_{h_t : \{x_{t'}\}_{t'=1}^{t-1} \cap \mathcal{X}_{l,3} \neq \emptyset} \pi_{1:t-1}T_{1:t-1}\mathbb{P}^*_{\mathcal{E}_{l,4}}(x_{1:t-1}, s_t) \|(D^{1,1}_{t-1:1} - D^3_{t-1:1})\mathbf{1}\|_1$$

$$\leq 2\epsilon_l \sum_{h_{t-1}:\{x_{t'}\}_{t'=1}^{t-2}\cap\mathcal{X}_{l,3}=\emptyset} \pi_{1:t-2}T_{1:t-2}\|(D_{t-2:1}^{1,1}-D_{t-2:1}^3)\mathbf{1}\|_1 \frac{\mathbb{P}_{\mathcal{E}_{l,4}}^*(x_{1:t-2},s_{t-1})}{m_{t-1}}$$

$$+ \sum_{h_{t-1}:\{x_{t'}\}_{t'=1}^{t-2}\cap\mathcal{X}_{l,3}\neq\emptyset} \pi_{1:t-2}T_{1:t-2}\|(D_{t-2:1}^{1,1}-D_{t-2:1}^3)\mathbf{1}\|_1\mathbb{P}_{\mathcal{E}_{l,4}}^*(x_{1:t-2},s_{t-1})$$

$$+ 2\epsilon_l \sum_{h_{t-1}} \pi_{1:t-2}T_{1:t-2}\|(D_+)_{t-2:1}\mathbf{1}\|_1\mathbb{P}_{\mathcal{E}_{l,4}}^*(x_{1:t-2},s_{t-1}).$$

The last term is less than $4\epsilon_l\mathbb{P}_{\mathcal{E}_{l,4}}^*(\phi)$, which proves equation (29).

## A.5    Analysis for Case II: $\mathcal{E}_{l,2}$ (equation (19))

For any $(x_i,x_j)\in\mathcal{X}_l$, let $\mathbb{P}^c(r_j|x_j,r_i,x_i):=\mu^1(x_i,x_j)/R_+(r_i|x_i)$ for $c=1,2$ (recall that we use $R_+(r|x):=\frac{1}{2}(R_1^1+R_2^1)(r|x)$, and $R_-^c(r|x):=\frac{1}{2}(R_1^c-R_2^c)(r|x)$). We first observe that

$$\|\mathbb{P}^1(r_j|x_j,r_i,x_i)-\mathbb{P}^2(r_j|x_j,r_i,x_i)\|_1 \leq O(\epsilon_l)/R_+(r_i|x_i), \tag{30}$$

since from the Bayes' rule, we have

$$\mathbb{P}^c(r_j|x_j,r_i,x_i) = \frac{R_1^c(r_i|x_i)}{R_1^c(r_i|x_i)+R_2^c(r_i|x_i)}R_1^c(r_j|x_j) + \frac{R_2^c(r_i|x_i)}{R_1^c(r_i|x_i)+R_2^c(r_i|x_i)}R_2^c(r_j|x_j),$$
$$= \frac{\mathbb{P}^c(r_i,r_j|x_i,x_j)}{R_+(r_i|x_i)},$$

for $c=1,2$. We start by writing the target errors as usual.

$$\sum_{\tau:x_{1:H}\in\mathcal{E}_{l,2}} |\mathbb{P}^{1,\pi}(\tau)-\mathbb{P}^{2,\pi}(\tau)| = \sum_{(x,r)_{1:H}} \mathbb{1}\{x_{1:H}\in\mathcal{E}_{l,2}\}T_{0:H-1}\pi_{1:H}\cdot|(\mathbb{P}^1-\mathbb{P}^2)(r_{1:H}|x_{1:H})|$$
$$= \sum_{h_H} \pi_{1:H-1}T_{0:H-1}\sum_{a_H,r_H} \mathbb{1}\{x_{1:H}\in\mathcal{E}_{l,2}\}\pi(a_H|h_H)\cdot|(\mathbb{P}^1-\mathbb{P}^2)(r_{1:H}|x_{1:H})|,$$

where we define

$$\mathbb{P}^c(r_{1:t}|x_{1:t}) := \frac{1}{2}\sum_{m=1}^2 \Pi_{t'=1}^t R_m^c(r_{t'}|x_{t'}).$$

for $c=1,2$ and any $t\in[H]$. Note that by Lemma A.3, we are guaranteed that all trajectories in $\mathcal{E}_{l,2}$ have at most 2 $\delta_l$-distinguishable state-actions in both models $\mathcal{M}^1,\mathcal{M}^2$. We split the cases when the number of distinguishable state-actions until time $H-1$ is whether $d=2,1,0$. Let $\mathcal{X}_{d,t}$ be a set of length $t$ transition histories such that the number of distinguishable state-actions is exactly $d$. That is,

$$\mathcal{X}_{d,t} = \left\{ x_{1:t-1}\Big| \sum_{t'=1}^{t-1}\mathbb{1}\{\Delta^1(x_{t'})>0 \text{ or } \Delta^2(x_{t'})>0\}=d \right\}. \tag{31}$$

With a slight abuse in notation, we overwrite the notation $\Delta(x):=\max(\Delta^1(x),\Delta^2(x))$. Note that when $\Delta(x)=0$ we have $D^1(\cdot|x)=D^2(\cdot|x)$ by construction of models $\mathcal{M}^1$ and $\mathcal{M}^2$.

We start from length $H$ histories $h_H$. We divide cases into whether $h_H$ belongs to $\mathcal{X}_{2,H}$, $\mathcal{X}_{1,H}$ or $\mathcal{X}_{0,H}$.

**Case i: $h_H : x_{1:H-1}\in\mathcal{X}_{2,H}$.**    Since we consider trajectories only in $\mathcal{E}_{l,2}$, we have $\Delta(x_H)=0$ and therefore the probability of observing $r_H$ under any history $h_H$ is the same, *i.e.*, $(\mathbb{P}^1-\mathbb{P}^2)(r_H|h_H)=0$. Hence we have

$$\sum_{a_H,r_H} \mathbb{1}\{x_{1:H}\in\mathcal{E}_{l,2}\}\pi(a_H|h_H)\cdot|(\mathbb{P}^1-\mathbb{P}^2)(r_{H:1}|x_{H:1})|$$
$$= \sum_{a_H} \mathbb{1}\{x_{1:H}\in\mathcal{E}_{l,2}\}\pi(a_H|h_H)\cdot|(\mathbb{P}^1-\mathbb{P}^2)(r_{H-1:1}|x_{H-1:1})|.$$

**Case ii:** $h_H : x_{1:H-1} \in \mathcal{X}_{1,H}$. We consider two cases: when $\Delta(x_H) = 0$ or $\Delta(x_H) > 0$. Let us define an event $\mathcal{E}_{d,t}$ which is defined as the following:

$$\mathcal{E}_{d,t} = \mathcal{E}_{l,2} \cap \{\tau = (x,r)_{1:H} | x_{1:t-1} \in \mathcal{X}_{d-1,t} \cap \Delta(x_t) > 0\}, \tag{32}$$

for $d = 2, 1$ and $t \in [H]$. Now in case $\Delta(x_H) = 0$, we again have $(\mathbb{P}^1 - \mathbb{P}^2)(r_H | h_H) = 0$. Therefore we have

$$\sum_{r_H} |(\mathbb{P}^1 - \mathbb{P}^2)(r_{H:1} | x_{H:1})| = |(\mathbb{P}^1 - \mathbb{P}^2)(r_{H-1:1} | x_{H-1:1})|.$$

In order to handle the latter case when $\Delta(x_H) > 0$, we first define $t^*$ and $p_t$ as the following:

$$t^* : t^* < t, \ s.t. \ \Delta(x_{t^*}) > 0,$$
$$p_t := \mathbb{P}^1(r_{t^*} | x_{t^*}) = \mathbb{P}^2(r_{t^*} | x_{t^*}) = R_+(r_{t^*} | x_{t_t^*}). \tag{33}$$

Note that with the above definition, for length $t$ history $h_t$ in any trajectories in $\mathcal{E}_{l,2}$, we have

$$\mathbb{P}^c(r_{t:1} | x_{t:1}) = \mathbb{P}^c(r_t | x_t, x_{t^*}, r_{t^*}) \cdot \mathbb{P}^c(r_{t-1:1} | x_{t-1:1}), \qquad c = 1, 2,$$

since for any $t' \neq t^*, t$, we have $\Delta^1(x_{t'}) = \Delta^2(x_{t'}) = 0$. Recall the inequality (30),

$$\sum_{r_H} \mathbb{1}\{x_{1:H} \in \mathcal{E}_{l,2}\} \pi(a_H | h_H) |(\mathbb{P}^1 - \mathbb{P}^2)(r_{H:1} | x_{H:1})|$$

$$\leq \sum_{r_H} \mathbb{1}\{x_{1:H} \in \mathcal{E}_{l,2}\} \pi(a_H | h_H) |(\mathbb{P}^1 - \mathbb{P}^2)(r_H | h_H)| \mathbb{P}^1(r_{1:H-1} | x_{1:H-1})$$

$$+ \sum_{r_H} \mathbb{1}\{x_{1:H} \in \mathcal{E}_{l,2}\} \pi(a_H | h_H) \mathbb{P}^2(r_H | h_H) |(\mathbb{P}^1 - \mathbb{P}^2)(r_{1:H-1} | x_{1:H-1})|$$

$$\leq \mathbb{1}\{x_{1:H} \in \mathcal{E}_{2,H}\} \pi(a_H | h_H) \left( \frac{\epsilon_l}{p_H} \mathbb{P}^1(r_{1:H-1} | x_{1:H-1}) + |(\mathbb{P}^1 - \mathbb{P}^2)(r_{1:H-1} | x_{1:H-1})| \right).$$

In the last inequality, since we are handling the special case when $x_{1:H} \in \mathcal{E}_{2,H}$, we replace $\mathbb{1}\{x_{1:H} \in \mathcal{E}_{l,2}\}$ by $\mathbb{1}\{x_{1:H} \in \mathcal{E}_{2,H}\}$.

**Case iii:** $h_H \in \mathcal{E}_{0,H}$. We first observe that $\mathbb{P}^1(r_{1:H-1} | x_{1:H-1}) = \mathbb{P}^2(r_{1:H-1} | x_{1:H-1})$. Furthermore, in this case $\mathbb{P}^1(r_H | h_H) = \mathbb{P}^1(r_H | x_H) = R_+(r_H | x_H)$. Thus

$$\sum_{a_H, r_H} \mathbb{1}\{x_{1:H} \in \mathcal{E}_{l,2}\} \pi(a_H | h_H) |(\mathbb{P}^1 - \mathbb{P}^2)(r_{H:1} | x_{H:1})| = 0.$$

**Combine all cases.** Plugging all of this into the summation over $h_H$:

$$\sum_{h_H} \pi_{1:H-1} T_{0:H-1} \sum_{a_H, r_H} \mathbb{1}\{x_{1:H} \in \mathcal{E}_{l,2}\} \pi(a_H | h_H) |(\mathbb{P}^1 - \mathbb{P}^2)(r_{H:1} | x_{H:1})|$$

$$\leq \sum_{h_H} \pi_{1:H-1} T_{0:H-1} |(\mathbb{P}^1 - \mathbb{P}^2)(r_{H-1:1} | x_{H-1:1})| \mathbb{P}^*_{\mathcal{E}_{l,2}}(h_H)$$

$$+ \sum_{h_H : x_{1:H-1} \in \mathcal{X}_{1,H}} \pi_{1:H-1} T_{0:H-1} \mathbb{P}^1(r_{H-1:1} | x_{H-1:1}) \frac{\epsilon_l}{p_H} \sum_{a_H} \mathbb{1}\{x_{1:H} \in \mathcal{E}_{2,H}\} \pi(a_H | h_H). \tag{34}$$

Finally, we control the last term in the above equation. We again consider the case when $\Delta(x_{H-1}) = 0$ and $\Delta(x_{H-1}) > 0$ which gives

$$\sum_{h_H : x_{1:H-1} \in \mathcal{X}_{1,H}} \pi_{1:H-1} T_{0:H-1} \mathbb{P}^1(r_{H-1:1} | x_{H-1:1}) \frac{\epsilon_l}{p_H} \sum_{a_H} \mathbb{P}^*_{\mathcal{E}_{2,H}}(x_{1:H}) \pi(a_H | h_H)$$

$$\leq \sum_{h_{H-1} : x_{1:H-2} \in \mathcal{X}_{1,H-1}} \pi_{1:H-2} T_{0:H-2} \mathbb{P}^1(r_{H-2:1} | x_{H-2:1}) \frac{\epsilon_l \mathbb{P}^*_{\mathcal{E}_{2,H}}(h_{H-1})}{p_{H-1}}$$

$$+ 2\epsilon_l \sum_{h_{H-1} : x_{1:H-2} \in \mathcal{X}_{0,H-1}} \pi_{1:H-2} T_{0:H-2} \mathbb{P}^1(r_{H-2:1} | x_{H-2:1}) \sum_{a_{H-1} : \Delta(x_{H-1}) > 0} \pi(a_{H-1} | h_{H-1}) \mathbb{P}^*_{\mathcal{E}_{2,H}}(x_{1:H-1})$$

$$\leq \sum_{h_{H-1}:x_{1:H-2}\in\mathcal{X}_{1,H-1}} \pi_{1:H-2}T_{0:H-2}\mathbb{P}^1(r_{H-2:1}|x_{H-2:1})\frac{\epsilon_l\mathbb{P}^*_{\mathcal{E}_{2,H}}(h_{H-1})}{p_{H-1}}$$

$$+ 2\epsilon_l \sum_{h_{H-1}:x_{1:H-2}\in\mathcal{X}_{0,H-1}} \sum_{a_{H-1}:\Delta(x_{H-1})>0} \mathbb{P}^{1,\pi}(h_{H-1},a_{H-1})\mathbb{P}^*_{\mathcal{E}_{2,H}}(x_{1:H-1}). \tag{35}$$

The first term comes from the fact that $p_{H-1} = p_H$ in case $\Delta(x_{H-1}) = 0$. For the second term, we note that since $x_{1:H-2} \in \mathcal{X}_{0,H-1}$, we have $\mathbb{P}^1(r_{H-2:1}|x_{H-2:1}) = (R_+)_{H-2:1}$, which is equivalently $\mathbb{P}^1(r_{H-1:1}|x_{H-1:1}) = (R_+)_{H-1}\mathbb{P}^1(r_{H-2:1}|x_{H-2:1})$. We apply this relation to obtain the above inequality (35).

We repeat this induction recursively until time step 1, and we get

$$\sum_{h_H:x_{1:H-1}\in\mathcal{X}_{1,H}} \pi_{1:H-1}T_{0:H-1}\mathbb{P}^1(r_{H-1:1}|x_{H-1:1})\frac{\epsilon_l\mathbb{P}^*_{\mathcal{E}_{2,H}}(h_H)}{p_H}$$

$$\leq 2\epsilon_l \sum_{t=1}^{H-1} \sum_{h_t:x_{1:t-1}\in\mathcal{X}_{0,t}} \sum_{a_t:\Delta(x_t)>0} \mathbb{P}^*_{\mathcal{E}_{2,H}}(x_{1:t})\mathbb{P}^\pi(h_t,a_t)$$

$$\leq 2\epsilon_l \sum_{t=1}^{H-1} \sum_{h_t:x_{1:t-1}\in\mathcal{X}_{0,t}} \sum_{a_t:\Delta(x_t)>0} \mathbb{P}^*_{\mathcal{E}_{1,t}}(x_{1:t})\mathbb{P}^\pi(h_t,a_t).$$

The last inequality follows from the fact that conditioned on $x_{1:t}$ such that $x_{1:t-1} \in \mathcal{X}_{0,t}$ and $\Delta(x_t) > 0$, we get $\mathbb{P}^*_{\mathcal{E}_{1,t}}(x_{1:t}) \geq \mathbb{P}^*_{\mathcal{E}_{2,H}}(x_{1:t})$.

Finally, we observe that $\mathcal{E}_{1,t} \cap \mathcal{E}_{1,t'} = \emptyset$ for all $t,t' \in [H]$ such that $t \neq t'$, and $\cup_{t=1}^H \mathcal{E}_{1,t} \subseteq \mathcal{E}_{l,2}$. Therefore there is no duplication of histories in the summation and we can safely proceed to:

$$\sup_\pi \sum_{t=1}^{H-1} \sum_{h_t:x_{1:t-1}\in\mathcal{X}_{0,t}} \sum_{a_t:\Delta(x_t)>0} \mathbb{P}^*_{\mathcal{E}_{1,t}}(h_t,a_t)\mathbb{P}^\pi(h_t,a_t)$$

$$= \sup_\pi \sum_{t=1}^{H-1} \sum_{(h_t,a_t):(x_{1:t-1}\in\mathcal{X}_{0,t})\cap\Delta(x_t)>0} \mathbb{P}^\pi(\mathcal{E}_{1,t}|h_t,a_t)\mathbb{P}^\pi(h_t,a_t)$$

$$\leq \sup_\pi \sum_{t=1}^{H-1} \sum_{(h_t,a_t):(x_{1:t-1}\in\mathcal{X}_{0,t})\cap\Delta(x_t)>0} \mathbb{P}^\pi(\mathcal{E}_{1,t},h_t,a_t) \leq \sup_\pi \sum_{t=1}^{H-1} \mathbb{P}^\pi(\mathcal{E}_{1,t}) \leq \sup_\pi \mathbb{P}^\pi(\mathcal{E}_{l,2}).$$

This concludes that

$$\sum_{h_H\in\mathcal{E}_{1,H}} \pi_{1:H-1}T_{0:H-1}\mathbb{P}^1(r_{H-1:1}|x_{H-1:1})\frac{\epsilon_l\mathbb{P}^*_{\mathcal{E}_{l,2}}(h_H)}{p_H} \leq 2\epsilon_l\mathbb{P}^*_{\mathcal{E}_{l,2}}(\phi),$$

and we can plug this back to (34) and apply the same argument recursively until time step drops down to 1. This gives

$$\sum_{\tau:x_{1:H}\in\mathcal{E}_{l,2}} |\mathbb{P}^{1,\pi}(\tau) - \mathbb{P}^{2,\pi}(\tau)| \leq O(H\epsilon_l)\mathbb{P}^*_{\mathcal{E}_{l,2}}(\phi).$$

---

**Algorithm 4** Reward Model Recovery from Second-Order Correlations

---

1: Solve an LP for $\hat{l}(x)$ for all $x \in \mathcal{S} \times \mathcal{A}$ such

$$\min_{\hat{l}(x):x\in\mathcal{S}\times\mathcal{A}} \quad \sum_{x\in\mathcal{S}\times\mathcal{A}} \hat{l}(x),$$

$$\text{s.t.} \quad \hat{l}(x) \text{ satisfy } (36), (37), \text{ and } (38) \tag{39}$$

2: Decide signs $sign(x)$ that satisfies (40).
3: Clip $\hat{l}(x)$ within $(-\infty, u(x)]$ where $u(x) = \log\left(\min(\hat{p}_+(x), 1 - \hat{p}_+(x))\right)$ for all $x$.
4: Set $\hat{R}_-(r = 1|x) = sign(x) \cdot \exp(\hat{l}(x))$ and $\hat{R}_-(r = 0|x) = -sign(x) \cdot \exp(\hat{l}(x))$ for all $x$.
5: Let $\hat{R}_1(r|x) = \hat{R}_+(r|x) + \hat{R}_-(r|x), \hat{R}_2(r|x) = \hat{R}_+(r|x) - \hat{R}_-(r|x)$ for all $r, x$.
6: Return $\hat{\mathcal{M}} = (\mathcal{S}, \mathcal{A}, \hat{T}, \hat{\nu}, \{\hat{R}_m\}_{m=1}^2)$.

---

# Appendix B  Deferred Analysis in Section 3

## B.1  Formulation of the LP for Model Recovery

We describe a detailed LP formulation to obtain an empirical reward models. Let $c_u(x_i, x_j)$ and $c_l(x_i, x_j)$ are upper and lower confidence bounds for $u(x_i, x_j)$ respectively:

$$c_u(x_i, x_j) := \hat{\mu}(x_i, x_j) - \hat{p}_+(x_i)\hat{p}_+(x_j) + \sqrt{\frac{\iota_2}{n(x_i, x_j)}} + \epsilon_0^2,$$

$$c_l(x_i, x_j) := \hat{\mu}(x_i, x_j) - \hat{p}_+(x_i)\hat{p}_+(x_j) - \sqrt{\frac{\iota_2}{n(x_i, x_j)}} - \epsilon_0^2.$$

We can write down the linear program to find variables $\hat{l}(x)$ with several linear constraints. The first constraint is upper bound on the multiplication of two reward differences:

$$\hat{l}(x_i) + \hat{l}(x_j) \le \log\left(|c_u(x_i, x_j)| \vee |c_l(x_i, x_j)|\right), \forall x_i, x_j \in \mathcal{S} \times \mathcal{A}, \tag{36}$$

The second constraint is, for all $x_i, x_j \in \mathcal{S} \times \mathcal{A}$ such that signs of both upper and lower bounds are equal, a lower confidence bound:

$$\hat{l}(x_i) + \hat{l}(x_j) \ge \log\left(|c_u(x_i, x_j)| \wedge |c_l(x_i, x_j)|\right), \ \forall(x_i, x_j): \ c_u(x_i, x_j) \cdot c_l(x_i, x_j) > 0. \tag{37}$$

Finally, we add a regularity condition for differences in probabilities:

$$10\log(\epsilon_0) \le \hat{l}(x) \le \log\left((\hat{p}_+(x) \wedge 1 - \hat{p}_+(x)) + \sqrt{\iota_1/n(x)} + \epsilon_0^2\right), \forall x \in \mathcal{S} \times \mathcal{A}. \tag{38}$$

with a properly set confidence parameter $\iota_1 = O(\log(SA/\eta))$. Now we can solve the LP feasibility problem for $\hat{l}(x)$ and find parameters for an empirical model. The LP formulation can be found in (39). The minimization objective $\sum \hat{l}(x)$ encourages to ignore small differences in two reward models at each state-action.

After we obtain a value for $l(x) = \log|\hat{\Delta}(x)|$, it remains to find signs of $\hat{p}_-(x)$. We find a correct assignment for $sign(x)$ with the following constraints:

$$sign(x_i)sign(x_j) = sign(\hat{u}(x_i, x_j)), \ \forall(x_i, x_j): \ c_u(x_i, x_j) \cdot c_l(x_i, x_j) > 0. \tag{40}$$

Solving constraints (40) can be formulated as 2-Satisfiability (2-SAT) problem [2] and thus can be efficiently solved. The full algorithm is described in Algorithm 4.

## B.2  Proof of Lemma 3.1

**Magnitude Constraints**  We start with confidence bounds for estimated probabilities that holds with probabilities at least $1 - \eta$:

$$|p_+(x) - \hat{p}_+(x)| \le \sqrt{\frac{c_1 \log(SA/\eta)}{n(x)}}, \qquad\qquad \forall x \in \mathcal{S} \times \mathcal{A},$$

$$\left|\mu(x_i, x_j) - \hat{\mu}(x_i, x_j)\right| \leq \sqrt{\frac{c_2 \log(SA/\eta)}{n(x_i, x_j)}}, \qquad\qquad \forall x_i, x_j \in \mathcal{S} \times \mathcal{A},$$

for some absolute constants $c_1, c_2$. Furthermore, by construction the number of visit counts for any pair is less than the visit counts of a single state-action:

$$\min(n(x_i), n(x_j)) \geq n(x_i, x_j), \qquad \forall x_i, x_j \in \mathcal{S} \times \mathcal{A}.$$

Then we consider a model $\mathcal{M}^1$ that approximates the true model with non-zero probability for every reward:

$$R_m^1(r|x) = R_m(r|x), \ m \in \{1, 2\}, \qquad\qquad r \in \{0, 1\}, x \in \mathcal{S} \times \mathcal{A} : |\Delta(x)| > \epsilon_0^5,$$

$$\begin{cases} R_1^1(r|x) = \epsilon_0^3 + (1 - 2\epsilon_0^3) R_1(r|x) + (2r - 1)\epsilon_0^4 \\ R_2^1(r|x) = \epsilon_0^3 + (1 - 2\epsilon_0^3) R_2(r|x) - (2r - 1)\epsilon_0^4 \end{cases}, \qquad r \in \{0, 1\}, x \in \mathcal{S} \times \mathcal{A} : |\Delta(x)| \leq \epsilon_0^5,$$

For this model, we have

$$|(R_+^1 - R_+)(r|x)| \leq \epsilon_0^3, \qquad\qquad \forall r \in \{0, 1\}, x \in \mathcal{S} \times \mathcal{A},$$

$$|(\mathbb{P}^1 - \mathbb{P})(r_i, r_j|x_i, x_j)| \leq 2\epsilon_0^3, \qquad\qquad \forall r_i, r_j \in \{0, 1\}, x_i, x_j \in \mathcal{S} \times \mathcal{A},$$

$$\Delta^1(x) \geq \epsilon_0^5, \qquad\qquad \forall x \in \mathcal{S} \times \mathcal{A}.$$

We show that one feasible solution $l^*(x)$ can be obtained from $\mathcal{M}^1$

$$l^*(x) := \log \left|p_-^1(x)\right| = \log(\Delta^1(x)).$$

We first check the first constraint (36):

$$\Delta^1(x_i) \Delta^1(x_j) = |\mu^1(x_i, x_j) - p_+^1(x_i) p_+^1(x_j)|$$

$$\leq |\hat{\mu}(x_i, x_j) - \hat{p}_+(x_i)\hat{p}_+(x_j)| + \sqrt{\frac{c_2 \log(SA/\eta)}{n(x_i, x_j)}} + \epsilon_0^2$$

$$= \max(|c_u(x_i, x_j)|, |c_l(x_i, x_j)|).$$

For the second constraint (37) follows similarly when $c_u(x_i, x_j) \cdot c_l(x_i, x_j) > 0$. In the other case, $\Delta^1(x_i)\Delta^1(x_j)$ can be as small as 0, which implies that $l(x_i) + l(x_j)$ is unbounded below. For the final constraint (38), we observe that

$$\epsilon_0^5 \leq \Delta^1(x) \leq p_+^1(x) \leq \hat{p}_+(x) + \sqrt{\frac{c_1 \log(SA/\eta)}{n(x)}} + \epsilon_0^3, \qquad\qquad \forall x \in \mathcal{S} \times \mathcal{A}.$$

Thus $l^*(x)$ satisfies all magnitude constraints (36), (37), and (38).

**Sign Constraint**  For the sign constraint (40), we note that whenever $c_u(x_i, x_j) \cdot c_l(x_i, x_j) > 0$, we have

$$sign(p_-^1(x_i) \cdot p_-^1(x_j)) = sign(x_i)sign(x_j) = sign(c_u(x_i, x_j)), \qquad\qquad (41)$$

which can be inferred by

$$p_-^1(x_i)p_-^1(x_2) = \mathbb{P}^1(r_i = 1, r_j = 1|x_i, x_j) - p_+^1(x_i)p_+^1(x_j)$$

$$= \hat{\mathbb{P}}(r_i = 1, r_j = 1|x_i, x_j) - \hat{p}_+(x_i)\hat{p}_+(x_j) + errors,$$

where $errors$ is in order of $O\left(\sqrt{c_2 \log(SA/\eta)/n(x_i, x_j)} + \epsilon^2\right)$. This implies that

$$c_l(x_i, x_j) \leq p_-^1(x_i)p_-^1(x_j) \leq c_u(x_i, x_j),$$

which concludes that reward models from $\mathcal{M}^1$ satisfy (40).

## B.3   Proof of Lemma 4.2

At this point, we no longer assume that the true transition and initial state models are given, but instead we use estimated quantities $\hat{T}, \hat{\nu}$ from Algorithm 2.

**Notations** Let us define a few variables to proceed. We denote $\#_k(x)$ as visit counts at $x$ during the $k^{th}$ episode. A similar quantity $\#_k(x_i, x_j)$ is 1 if data for a pair $(x_i, x_j)$ is collected in the $k^{th}$ episode and 0 otherwise. Let $\widetilde{\pi}_k$ be the policy executed in the $k^{th}$ episode. Let $n_k(x) := \sum_{k'=1}^{k} \#_{k'}(x)$, $n_k(x_i, x_j) = \sum_{k'=1}^{k} \#_{k'}(x_i, x_j)$ and the expected quantities $\bar{n}_k(x) = \sum_{k'=1}^{k} \mathbb{E}^{\widetilde{\pi}_{k'}}[\#_{k'}(x)]$, $\bar{n}_{k'=1}^{k}(x_i, x_j) = \sum_{k'=1}^{k} \mathbb{E}^{\widetilde{\pi}_{k'}}[\#_{k'}(x_i, x_j)]$. We define a desired high probability event $\mathcal{E}_{pe}$ for martingale sums:

$$n_k(x) \geq \frac{1}{2} \bar{n}_k(x) - c_l \log(K/\eta), \qquad \forall k \in [K], x \in \mathcal{S} \times \mathcal{A},$$

$$n_k(x_i, x_j) \geq \frac{1}{2} \bar{n}_k(x_i, x_j) - c_l \log(K/\eta), \qquad \forall k \in [K], x_i, x_j \in \mathcal{S} \times \mathcal{A}, \tag{42}$$

for some absolute constant $c_l > 0$. With a standard measure of concentration argument for martingale sums [48], we can show that $\mathbb{P}(\mathcal{E}_{pe}) \geq 1 - \eta$. We also denote $\hat{T}_k, \hat{\nu}_k$ for the empirically estimated transition and initial distribution models at the beginning of $k^{th}$ episode. Let $v_1$ be a 2-tuple $(null, null)$ and $i_1 = 1$.

**Number of Episodes $K$** We first show that we terminate Algorithm 2 after at most $K$ episodes with probability at least $1 - \eta$ where

$$K = C \cdot \frac{S^2 A}{\epsilon_{pe}^2} (H + A) \log(K/\eta),$$

for some absolute constant $C > 0$. Let us examine $\widetilde{V}_0$ at the $k^{th}$ episode. This can be decomposed as

$$\widetilde{V}_0 = \sqrt{\iota_\nu/k} + \sum_s \hat{\nu}_k(s) \cdot \widetilde{V}_1(i_1, v_1, s)$$

$$\leq \sqrt{\iota_\nu/k} + \|\hat{\nu}_k(s) - \nu(s)\|_1 + \sum_s \nu(s) \cdot \widetilde{Q}_1((i_1, v_1, s), (a, z))$$

$$\leq 2\sqrt{\iota_\nu/k} + \mathbb{E}^{\widetilde{\pi}_k}\left[\widetilde{Q}_1((i_1, v_1, s_1), \widetilde{\pi}_k(v_1, s_1))\right].$$

Note that $(a_t, z_t) = \widetilde{\pi}_k(v_t, s_t)$ for $t \in [H]$. We can recursively bound expectation of $\widetilde{Q}_t$ for $t \geq 1$:

$$\mathbb{E}^{\widetilde{\pi}_k}\left[\widetilde{Q}_t((i_t, v_t, s_t), (a_t, z_t))\right]$$

$$= \mathbb{E}^{\widetilde{\pi}_k}[b_r(i_t, v_t, z_t) + b_T(s_t, a_t)] + \mathbb{E}^{\widetilde{\pi}_k}\left[\sum_{s_{t+1}} \hat{T}_k(s_{t+1}|s_t, a_t) \cdot \widetilde{V}_{t+1}(i_{t+1}, v_{t+1}, s_{t+1})\right]$$

$$\leq \mathbb{E}^{\widetilde{\pi}_k}\left[\sqrt{\frac{\iota_2 \mathbb{1}\{collect\}}{n_k(v_{t+1})}} + 2\sqrt{\frac{\iota_T}{n_k(s_t, a_t)}}\right] + \mathbb{E}^{\widetilde{\pi}_k}\left[\widetilde{Q}_{t+1}((i_{t+1}, v_{t+1}, s_{t+1}), (a_{t+1}, z_{t+1}))\right],$$

where $\mathbb{1}\{collect\}$ is a short hand for $\mathbb{1}\{i_t = 2 \cap z_t = 1\}$. We summarize all appended terms to get

$$\widetilde{V}_0 \leq 2\sqrt{\iota_\nu/k} + \sum_{t=1}^{H} \mathbb{E}^{\widetilde{\pi}_k}\left[2\sqrt{\frac{\iota_T}{n_k(s_t, a_t)}} + \sqrt{\frac{\iota_2 \mathbb{1}\{collect\}}{n_k(v_{t+1})}}\right]$$

$$\leq 2\sqrt{\iota_\nu/k} + \sum_{(s,a)} 2\sqrt{\frac{\iota_T}{n_k(s, a)}} \cdot \mathbb{E}^{\widetilde{\pi}_k}[\#_k(s, a)] + \sum_{(x_i, x_j)} \sqrt{\frac{\iota_2}{n_k(x_i, x_j)}} \cdot \mathbb{E}^{\widetilde{\pi}_k}[\#_k(x_i, x_j)]$$

$$\leq 2\sqrt{\iota_\nu/k} + \sum_x 2H \cdot \mathbb{1}\{\bar{n}_k(x) < 4 \cdot c_l \log(K/\eta)\}$$

$$+ 4\sum_x \sqrt{\frac{\iota_T}{\bar{n}_k(x)}} \cdot (\bar{n}_{k+1}(x) - \bar{n}_k(x)) + 4 \sum_{(x_i, x_j)} \sqrt{\frac{\iota_2}{\bar{n}_k(x_i, x_j)}} \cdot (\bar{n}_{k+1}(x_i, x_j) - \bar{n}_k(x_i, x_j)).$$

We now sum over $K$ episodes, then

$$K\epsilon_{pe} \leq 4\sqrt{\iota_\nu K} + O(c_l HSA \log(KSA/\eta)) + 8\sum_x \sqrt{\iota_T \bar{n}_{K+1}(x)} + 8 \sum_{(x_i, x_j)} \sqrt{\iota_2 \bar{n}_{K+1}(x_i, x_j)}.$$

We now note that $\sum_x \bar{n}_{K+1}(x) = HK$ and $\sum_{(x_i, x_j)} \bar{n}_{K+1}(x_i, x_j) \leq K$. Using a Cauchy-Schwartz inequality, we get

$$K\epsilon_{pe} \leq O\left(\sqrt{\iota_\nu K} + HSA\log(SA/\eta) + \sqrt{\iota_T HSAK} + \sqrt{\iota_2 S^2 A^2 K}\right).$$

Note we plug our design of confidence interval parameters $\iota_\nu = O(S\log(K/\eta))$, $\iota_T = O(S\log(KSA/\eta))$ and $\iota_2 = O(\log(KSA/\eta))$. This gives a bound that $K \leq O\left(S^2 A(A+H)\epsilon_{pe}^{-2}\log(KSA/\eta)\right)$. Hence we terminate Algorithm 2 after at most $K$ episodes under the event $\mathcal{E}_{pe}$

$$K = C \cdot S^2 A(A+H)\epsilon_{pe}^{-2}\log(HSA/(\epsilon_{pe}\eta)).$$

Note that from a concentration of martingale sums, $\mathcal{E}_{pe}$ happens with probability at least $1 - \eta$.

**Bound on $\mathbb{P}^*_{\mathcal{E}_l'}(\phi)$** We now show the second part of the lemma: under the event $\mathcal{E}_{pe}$, we have

$$\mathbb{P}^*_{\mathcal{E}_l'}(\phi) \leq O(H\epsilon_{pe}) \cdot \max\left(1, \sqrt{n_l/\iota_2}\right).$$

We first observe that

$$\mathbb{P}^*_{\mathcal{E}_l'} = \sup_\pi \mathbb{P}^\pi(\mathcal{E}_l') \leq \sup_\pi \mathbb{P}^\pi\left(\cup_{t_1=1}^{H-1}\{\exists t_2 > t_1,\ s.t.\ (x_{t_1}, x_{t_2}) \in \mathcal{X}_l \cap \mathcal{X}_{l-1}^c\}\right)$$

$$\leq \sup_\pi \sum_{t_1=1}^{H-1} \mathbb{P}^\pi\left(\exists t_2 > t_1,\ s.t.\ (x_{t_1}, x_{t_2}) \in \mathcal{X}_l \cap \mathcal{X}_{l-1}^c\right).$$

Now in the augmented MDP $\widetilde{\mathcal{M}}$ consider a class of augmented policies $\widetilde{\Pi}_{t_1, l}$ such that for a fixed $t_1$, a policy $\widetilde{\pi}$ picks the first state of a pair only at time step $t_1$, *i.e.*, $z_t = 0$ for $t < t_1$ and $z_{t_1} = 1$, and then pick the second state of the pair whenever it encounters that makes the pair belong to $\mathcal{X}_l$, *i.e.*, we set $z_{t_2} = 1$ whenever $(x_{t_1}, x_{t_2}) \in \mathcal{X}_l$ for $t_2 > t_1$. Within this policy class, define $\widetilde{Q}^1$ and $\widetilde{V}^1$ as the following:

$$b_T^1(s, a) = \left(1 \wedge \sqrt{\frac{\iota_T}{n(s, a)}}\right), \quad b_r^1(i, v, z) = \mathbb{1}\left\{i = 2 \cap z = 1 \cap (v' \in \mathcal{X}_l \cap \mathcal{X}_{l-1}^c)\right\} \cdot \left(1 \wedge \sqrt{\frac{\iota_2}{n_l}}\right),$$

$$\widetilde{Q}_t^1((i, v, s), (a, z)) = 1 \wedge \left(b_r + \sum_{s'} \hat{T}(s'|s, a) \cdot \widetilde{V}_{t+1}^1(i', v', s') + b_T\right),$$

with $\widetilde{Q}_{H+1}^1 = 0$. For $\widetilde{V}^1$, we define it as,

$$\widetilde{V}_t^1(i, v, s) = \max_{(a, z) \in \widetilde{\mathcal{A}}} \widetilde{Q}_t^1((i, v, s), (a, z)), \qquad \text{if } t > t_1,$$

$$\widetilde{V}_t^1(i, v, s) = \max_{a \in \mathcal{A}} \widetilde{Q}_t^1((i, v, s), (a, 1)), \qquad \text{if } t = t_1,$$

$$\widetilde{V}_t^1(i, v, s) = \max_{a \in \mathcal{A}} \widetilde{Q}_t^1((i, v, s), (a, 0)), \qquad \text{if } t < t_1,$$

and $\widetilde{V}_0^1 = \sqrt{\iota_\nu/K} + \sum_s \nu(s) \cdot \widetilde{V}_1^1(1, v_1, s)$ and $v_1 = (null, null)$. By construction, $\widetilde{V}_0$ is an upper confidence bound of $\widetilde{V}_0^1$:

$$\epsilon_{pe} \geq \widetilde{V}_0 \geq \widetilde{V}_0^1.$$

On the other hand, $\sup_\pi \mathbb{P}^\pi\left(\exists t_2 > t_1,\ s.t.\ (x_{t_1}, x_{t_2}) \in \mathcal{X}_l \cap \mathcal{X}_{l-1}^c\right)$ can be computed through the dynamic programming on $\widetilde{Q}^*$:

$$b_r(i, v, z) = \mathbb{1}\left\{i = 2 \cap z = 1 \cap (v' \in \mathcal{X}_l \cap \mathcal{X}_{l-1}^c)\right\} \cdot \left(1 \wedge \sqrt{\frac{\iota_2}{n_l}}\right),$$

$$\widetilde{Q}_t^*((i, v, s), (a, z)) = b_r + \sum_{s'} T(s'|s, a) \cdot \widetilde{V}_{t+1}^*(i', v', s'),$$

and

$$\widetilde{V}_t^*(i, v, s) = \max_{(a,z)\in\widetilde{\mathcal{A}}} \widetilde{Q}_t^*((i,v,s),(a,z)), \qquad\qquad \text{if } t > t_1,$$

$$\widetilde{V}_t^*(i, v, s) = \max_{a\in\mathcal{A}} \widetilde{Q}_t^*((i,v,s),(a,1)), \qquad\qquad \text{if } t = t_1,$$

$$\widetilde{V}_t^*(i, v, s) = \max_{a\in\mathcal{A}} \widetilde{Q}_t^*((i,v,s),(a,0)), \qquad\qquad \text{if } t < t_1,$$

Then the maximum probability of having $(x_{t_1}, x_{t_2}) \in \mathcal{X}_l \cap \mathcal{X}_{l-1}^c$ can be computed as the following:

$$\widetilde{V}_0^* = \sum_s \nu(s) \cdot \widetilde{V}^*(1, v_1, s) = \left(1 \wedge \sqrt{\frac{\iota_2}{n_l}}\right) \cdot \sup_\pi \mathbb{P}^\pi \left(\exists t_2 > t_1, \ s.t. \ (x_{t_1}, x_{t_2}) \in \mathcal{X}_l \cap \mathcal{X}_{l-1}^c\right).$$

Finally, we can inductively show that $\widetilde{Q}_t^1 \geq \widetilde{Q}_t^*$ and $\widetilde{V}_0^1 \geq \widetilde{V}_0^*$ with the setting of confidence interval parameters $\iota_2, \iota_T$. This concludes that

$$\widetilde{V}_0^1 \geq \widetilde{V}_0^* = \left(1 \wedge \sqrt{\frac{\iota_2}{n_l}}\right) \cdot \sup_\pi \mathbb{P}^\pi \left(\exists t_2 > t_1, \ s.t. \ (x_{t_1}, x_{t_2}) \in \mathcal{X}_l \cap \mathcal{X}_{l-1}^c\right).$$

This concludes that

$$\mathbb{P}_{\mathcal{E}_l'}^* \leq \sum_{t_1=1}^{H-1} \sup_\pi \mathbb{P}^\pi \left(\exists t_2 > t_1, \ s.t. \ (x_{t_1}, x_{t_2}) \in \mathcal{X}_l \cap \mathcal{X}_{l-1}^c\right) \leq H\epsilon_{pe} \cdot \left(1 \vee \sqrt{n_l/\iota_2}\right).$$

## B.4 Complete Proof of Theorem 3.2

In this section, we put the final puzzle piece, a guarantee on total-variation distance between trajectories with different transitions and initial state probabilities, to complete the picture. Let $\mathcal{M}^1$ be a 2RM-MDP model such that

$$\nu^1(s) = \nu(s), \ T^1(\cdot|s,a) = T(\cdot|s,a), \ R_m^1(\cdot|s,a) = \hat{R}(\cdot|s,a), \quad \forall m \in \{1,2\}, s \in \mathcal{S}, a \in \mathcal{A}$$

For any history-dependent policy $\pi$, we target to bound that

$$\sum_\tau |\mathbb{P}^\pi(\tau) - \hat{\mathbb{P}}^\pi(\tau)| \leq \sum_\tau |\mathbb{P}^\pi(\tau) - \mathbb{P}^{1,\pi}(\tau)| + |\mathbb{P}^{1,\pi}(\tau) - \hat{\mathbb{P}}^\pi(\tau)|.$$

Note that we already have shown that $\sum_\tau |\mathbb{P}^\pi(\tau) - \mathbb{P}^{1,\pi}(\tau)| \leq \epsilon/H$ in Section 4 and Appendix A.3. Thus we focus on bounding the second term. We first need the following lemma which is proven in Appendix D.

**Lemma B.1** *For any history-dependent policy $\pi$, we have*

$$\sum_\tau |\mathbb{P}^{1,\pi}(\tau) - \hat{\mathbb{P}}^\pi(\tau)| \leq \|(\nu - \hat{\nu})(s_1)\|_1 + \sum_{t=1}^{H-1} \mathbb{E}^{1,\pi}\left[\|(T - \hat{T})(s_{t+1}|s_t, a_t)\|_1\right]. \tag{43}$$

After $K$ episodes of pure-exploration, we also get

$$\|(\nu - \hat{\nu})(s_1)\|_1 + \sum_{t=1}^{H-1} \mathbb{E}^{1,\pi}\left[\|(T - \hat{T})(s_{t+1}|s_t, a_t)\|_1\right] \leq \sqrt{\iota_\nu/K} + \sum_{t=1}^{H-1} \mathbb{E}^{1,\pi}\left[\sqrt{\iota_T/n(s_t, a_t)}\right].$$

We can show that (43) is at most $2H\widetilde{V}_0$ (recall (5)) from the Bellman-update rule. First we see $\|(\nu - \hat{\nu})(s_1)\|_1 + \sum_{t=1}^{H-1} \mathbb{E}^{1,\pi}\left[\|(T - \hat{T})(s_{t+1}|s_t, a_t)\|_1\right] \leq \widetilde{V}_0^1$ where:

$$\widetilde{Q}_t^1(s, a) = \|(T - \hat{T})(s_{t+1}|s_t, a_t)\|_1 + \sum_{s'} T(s'|s,a) \cdot \widetilde{V}_{t+1}^1(s')$$

$$\leq (H + 1) \cdot \sqrt{\frac{\iota_T}{n(s_t, a_t)}} + \sum_{s'} \hat{T}(s'|s,a) \cdot \widetilde{V}_{t+1}^1(s'),$$

$$\widetilde{V}_t^1(s) = \max_a \widetilde{Q}_t^1(s,a),$$

$$\widetilde{V}_0^1 = \|(\nu - \hat{\nu})(s_1)\|_1 + \sum_s \nu(s) \cdot \widetilde{V}_1^1(s) \le (H+1) \cdot \sqrt{\frac{\iota_\nu}{K}} + \sum_s \hat{\nu}(s) \cdot \widetilde{V}_1^1(s),$$

and $\widetilde{V}_{H+1}^1 = 0$. On the other hand, from the construction of $\widetilde{Q}$ in (5), for any $i, v, z$, we have $2H \cdot \widetilde{Q}_t((i,v,s),(a,z)) \ge \widetilde{Q}_t^1(s,a)$ for all $(s,a)$. Therefore,

$$\sqrt{\iota_\nu/K} + \sum_{t=1}^{H-1} \mathbb{E}^\pi \left[ \sqrt{\iota_T/n(x_t)} \right] \le 2H\widetilde{V}_0 \le 2H\epsilon_{pe}. \tag{44}$$

With our choice of $\epsilon_{pe} = o(\epsilon/H^3)$, we have $\|\mathbb{P}^\pi(\tau) - \hat{\mathbb{P}}^\pi(\tau)\|_1 \le O(\epsilon/H)$, which then we can conclude that $|V_{\mathcal{M}}^\pi - V_{\hat{\mathcal{M}}}^\pi| \le H \cdot \|\mathbb{P}^\pi(\tau) - \hat{\mathbb{P}}^\pi(\tau)\|_1 \le O(\epsilon)$.

## Appendix C    Proof of the Lower Bound (Theorem 3.3)

We first consider instances of 2RM-MDP with $H = 2, S = 2$ and arbitrary $A \ge 3$. Our construction is as the following:

1. Regardless of actions played, both MDPs always start from $s_1$ at $t = 1$ and transits to $s_2$ at $t = 2$.

2. At $t = 1$, for all actions $a \in \mathcal{A}$ except $a_1^*$, both MDPs return a reward sampled from a Bernoulli distribution $Ber(1/2)$.
   - $\mathcal{M}_1$: For the action $a_1^*$, a reward is sampled from $Ber(1/2 + \sqrt{\epsilon})$.
   - $\mathcal{M}_2$: For the action $a_1^*$, a reward is sampled from $Ber(1/2 - \sqrt{\epsilon})$.

3. At $t = 2$, for all actions $a \in \mathcal{A}$ except $a_{2,1}^*$ and $a_{2,2}^*$, both MDPs return a reward sampled from a Bernoulli distribution $Ber(1/2)$.
   - $\mathcal{M}_1$: For $a_{2,1}^*, a_{2,2}^*$, rewards are sampled from $Ber(1/2 + \sqrt{\epsilon})$ and $Ber(1/2 - \sqrt{\epsilon})$ respectively.
   - $\mathcal{M}_2$: For $a_{2,1}^*, a_{2,2}^*$, rewards are sampled from $Ber(1/2 - \sqrt{\epsilon})$ and $Ber(1/2 + \sqrt{\epsilon})$ respectively.

In this construction, the optimal strategy is to play $a_1^*$ at $t = 1$, and depending on the outcome $r_1$, we play either $a_{2,1}^*$ if $r_1 = 1$ or $a_{2,2}^*$ otherwise. By playing this strategy, expected long-term return is

$$\frac{1}{2} \sum_{m=1}^2 \mathbb{P}_m(r_1 = 1) + \mathbb{P}_m(r_2 = 1|a_{2,1}^*) \cdot \mathbb{P}_m(r_1 = 1) + \mathbb{P}_m(r_2 = 1|a_{2,2}^*) \cdot \mathbb{P}_m(r_1 = 0)$$

$$= \frac{1}{2} + \left( (1/2 + \sqrt{\epsilon})^2 + (1/2 - \sqrt{\epsilon})^2 \right) = 1 + 2\epsilon.$$

Note that if we do not play $a_1^*$ at $t = 1$, or one of $a_{2,1}^*$ and $a_{2,2}^*$ at $t = 2$, then the expected cumulative reward is 1. Therefore the problem can be reduced to find $a_1^*$ and $a_{2,1}^*$ or $a_{2,2}^*$.

However, if we play $a_1 \neq a_1^*$ at $t = 1$ or play $a_2 \neq a_{2,1}^*, a_{2,2}^*$ at $t = 2$, *i.e.*, if we do not play the right actions at both time steps, then marginal distribution of the sample we get from the environment is always $\mathbb{P}(r_1, r_2|a_1, a_2) = \frac{1}{4}$ for all $(r_1, r_2) \in \{0,1\}^{\otimes 2}$. Therefore, even if we can access to a full distribution of outcomes from a wrong action sequence, there would be no information gain from playing any wrong sequences of actions, other than removing the played sequence from all $O(A^2)$ possibilities.

On the other hand, even if we play the correct sequence $(a_1^*, a_{2,1}^*)$ (or $(a_1^*, a_{2,2}^*)$), unless we play it sufficient number of times $O(1/\epsilon^2)$, we cannot distinguish it from other wrong action sequences. That is, the marginal distribution of the reward sequence from the correct action-sequence is close to the one obtained with any wrong action sequence $(a_1, a_2)$ in Kullback-Leibler (KL) divergence:

$$\sum_{(r_1, r_2) \in \{0,1\}^{\otimes 2}} \mathbb{P}(r_1, r_2|a_1, a_2) \cdot \log \left( \frac{\mathbb{P}(r_1, r_2|a_1, a_2)}{\mathbb{P}(r_1, r_2|a_1^*, a_{2,1}^*)} \right) < O(\epsilon^2).$$

We can apply a similar lower-bound argument for the multi-armed bandits with $A^2$-arms in [21], where a general framework for the lower-bound is provided based on information gains from the played policies. Note that there are always $A^2$-possible sequences of actions to play in each episode, and the information gain after each episode is not affected by how the decision in each step is made. This argument gives a $\Omega(A^2/\epsilon^2)$ lower bound.

Then following the action-amplification argument (see Appendix A in [36]), we can obtain $\Omega(S^2 A^2/\epsilon^2)$ lower bound.

## Appendix D    Auxiliary Lemmas

### D.1    Proof of Lemma 4.3

Recall that $l(x_i) + l(x_j) = \log(\Delta(x_i)) + \log(\Delta(x_j)) = \log|u(x_i, x_j)|$ for all $x_i, x_j \in \mathcal{S} \times \mathcal{A}$ when exact model parameters are given from equation (6). Thus if we have three equations for $(x_1, x_2), (x_2, x_3), (x_3, x_1)$, then we can compute $l(x_1)$ as follows:

$$l(x_1) = \frac{1}{2}(\log|u(x_1, x_2)| + \log|u(x_2, x_3)| + \log|u(x_3, x_1)|) - \log|u(x_2, x_3)|.$$

Now from empirical estimates of the model $\hat{\mathcal{M}}$, we can compute $\hat{l}(x_1)$ similarly. To simplify the burden in notation, let us denote $z_1 = u(x_1, x_2)$, $z_2 = u(x_2, x_3)$ and $z_3 = u(x_3, x_1)$, and empirical counterparts as $\hat{z}_1, \hat{z}_2, \hat{z}_3$. We first note that

$$\begin{aligned}
|z_1 - \hat{z}_1| &= |u(x_1, x_2) - \hat{u}(x_1, x_2)| \\
&\leq |\mu(x_1, x_2) - \hat{\mu}(x_1, x_2)| + |p_+(x_1)p_+(x_2) - \hat{p}_+(x_1)\hat{p}_+(x_2)| + b(x_1, x_2) \\
&\leq 3\sqrt{\iota_2/n(x_1, x_2)} + \epsilon_0^2 \leq 0.01\delta_l\epsilon_l,
\end{aligned}$$

where the last inequality uses $n(x_1, x_2) \geq n_l = C \cdot \iota_2\delta_l^{-2}\epsilon_l^{-2}$ for a sufficiently large $C > 0$. Similarly, for $z_2 - \hat{z}_2$ and $z_3 - \hat{z}_3$ we can obtain similar inequalities. On the other hand, we note that $|z_1| = \Delta(x_1)\Delta(x_2) \geq \delta_l^2 \gg 0.01\delta_l\epsilon_l$. Now we can see that

$$\begin{aligned}
|l(x_1) - \hat{l}(x_1)| &\leq \frac{1}{2}\left(\frac{0.01\delta_l\epsilon_l}{z_1} + \frac{0.01\delta_l\epsilon_l}{z_2} + \frac{0.01\delta_l\epsilon_l}{z_3}\right) \\
&\leq \frac{1}{2}\left(\frac{0.01\delta_l\epsilon_l}{\Delta(x_1)\Delta(x_2)} + \frac{0.01\delta_l\epsilon_l}{\Delta(x_2)\Delta(x_3)} + \frac{0.01\delta_l\epsilon_l}{\Delta(x_3)\Delta(x_1)}\right).
\end{aligned}$$

Let $i^* = \arg\max_{i \in [3]} \Delta(x_i) = 2$ (the same argument holds for any $i^*$ by symmetry). First observe that

$$|l(x_1) - \hat{l}(x_1)| \leq \frac{0.01\epsilon_l}{2}\left(\frac{1}{\Delta(x_1)} + \frac{2}{\Delta(x_2)}\right),$$

where we used $\Delta(x_2), \Delta(x_3) \geq \delta_l$. Now converting $l(x)$ to $\Delta(x_1) = \exp(l(x_1))$, we get

$$\begin{aligned}
\hat{\Delta}(x_1) &\leq \Delta(x_1)\exp\left(\frac{0.01\epsilon_l}{2}\left(\frac{1}{\Delta(x_1)} + \frac{2}{\Delta(x_2)}\right)\right) \\
&\leq \Delta(x_1)\left(1 + 2 \cdot \frac{0.01\epsilon_l}{2}\left(\frac{1}{\Delta(x_1)} + \frac{2}{\Delta(x_2)}\right)\right) \\
&\leq \Delta(x_1) + 0.03\epsilon_l,
\end{aligned}$$

where in the second inequality we used $\exp(z) \leq 1 + 2z$ for small enough $z$. Similarly, we can also show $\hat{\Delta}(x_1) \geq \Delta(x_1) - 0.03\epsilon_l$. Symmetric argument holds for $\Delta(x_3)$. For $\Delta(x_2)$, let $i_2^* = \arg\max_{i \in \{1,3\}} \Delta(x_i)$. Then similarly we get

$$\begin{aligned}
\hat{\Delta}(x_2) &\leq \Delta(x_2)\left(1 + \frac{0.01\epsilon_l}{2}\left(\frac{2}{\Delta(x_2)} + \frac{1}{\Delta(x_{i_2^*})}\right)\right) \\
&\leq \Delta(x_2) + 0.03\epsilon_l \frac{\Delta(x_2)}{\Delta(x_{i_2^*})}.
\end{aligned}$$

This concludes the proof of Lemma 4.3.

## D.2 Proof of Corollary 1

By Lemma 4.3, for any $\tau = (x, r)_{1:H} \in \mathcal{E}_{l,3}$ and $t \neq t^*$, we have that $\Delta(x_t) \geq \delta_l$ implies $|\Delta(x_t) - \hat{\Delta}(x_t)| \leq \epsilon_l/2$. The bound for $x_{t^*}$ also directly follows from Lemma 4.3.

For $\Delta(x_t) < \delta_l$, if $\delta_l = \max(\delta, \epsilon_l) = \epsilon_l$, then we have $\Delta(x_{t^*})\Delta(x_t) < \epsilon_l\Delta(x_{t^*})$. Now by the constraint (7), we have

$$|p_-(x_{t^*})p_-(x_t) - \hat{p}_-(x_{t^*})\hat{p}_-(x_t)| < 0.01\delta_l\epsilon_l.$$

From this, we have $\hat{\Delta}(x_{t^*})\hat{\Delta}(x_t) < 1.01\epsilon_l\Delta(x_{t^*})$. At the same time, by Lemma 4.3 we also have $\hat{\Delta}(x_{t^*}) \geq 0.5\Delta(x_t^*)$. Therefore, we get $\hat{\Delta}(x_t) < 3\epsilon_l$.

If $\Delta(x_t) < \delta_l$ and $\delta_l = \max(\delta, \epsilon_l) = \delta > \epsilon_l$, then by definition of $\delta$, we have $\Delta(x_t) = 0$. In this case, from $|\hat{\Delta}(x_{t^*})\hat{\Delta}(x_t) - \Delta(x_t)\Delta(x_t)| < \sqrt{\iota_2/n_l} < 0.03\epsilon_l\delta$. From this, we get $\hat{\Delta}(x_{t^*})\hat{\Delta}(x_t) < 0.03\epsilon_l\delta$. Since $\hat{\Delta}(x_{t^*}) \geq 0.5\Delta(x_t^*) \geq 0.5\delta$, we get $\hat{\Delta}(x_t) < 2\epsilon_l$.

## D.3 Proof of Lemma A.3

We start with the following inequality:

$$\begin{aligned}
&|u(x_1, x_2) - \hat{u}(x_1, x_2)| \\
&\leq |\mu(x_1, x_2) - \hat{\mu}(x_1, x_2)| + |p_+(x_1)p_+(x_2) - \hat{p}_+(x_1)\hat{p}_+(x_2)| + b(x_1, x_2) \\
&\leq 3\sqrt{\iota_2/n(x_1, x_2)} + \epsilon_0^2 \leq 0.01\delta_l\epsilon_l
\end{aligned} \tag{45}$$

for any $x_1, x_2$ such that $n(x_1, x_2) \geq n_l$.

**Case I:** $\sum_{t=1}^{H} \mathbb{1}\{\Delta(x_t) \geq \epsilon_l\} = 2$. In this case, let any time-step $t_1$ such that $\hat{\Delta}(x_{t_1}) \geq 2\epsilon_l$, and we show that $\Delta(x_{t_1}) < \epsilon_l$. If it were true, let $t_2, t_3$ be the time steps where $\Delta(x_t) \geq \epsilon_l$ for $t = t_2, t_3$. Then,

$$\hat{\Delta}(x_t) < \Delta(x_t)/1.5, \qquad t = t_2, t_3. \tag{46}$$

If this is the case, then $|\hat{u}(x_{t_2}, x_{t_3}) - u(x_{t_2}, x_{t_3})| > u(x_{t_2}, x_{t_3})/2 > \delta_l\epsilon_l/2$. However, visit counts for all pairs in a same trajectory satisfies $n(x_{t_2}, x_{t_3}) \geq C \cdot \iota_2\delta_l^{-2}\epsilon_l^{-2}$, and thus $|\hat{u}(x_{t_2}, x_{t_3}) - u(x_{t_2}, x_{t_3})| < 0.01\delta_l\epsilon_l$, which is contradiction (see equation (45)).

Now we argue that (46) is true. Suppose for $t_2$, we have $\hat{\Delta}(x_{t_2}) \geq \Delta(x_{t_2})/1.5$. Then,

$$\hat{u}(x_{t_1}, x_{t_2}) = \hat{\Delta}(x_{t_1})\hat{\Delta}(x_{t_2}) > \frac{4}{3}\epsilon_l\Delta(x_{t_2}).$$

On the other hand, we have $u(x_{t_1}, x_{t_2}) < \epsilon_l\Delta(x_{t_2})$. Now we can see that

$$|\hat{u}(x_{t_1}, x_{t_2}) - u(x_{t_1}, x_{t_2})| \geq \frac{1}{3}\epsilon_l\Delta(x_{t_2}) > \frac{1}{3}\epsilon_l\delta_l,$$

which is again contradiction since $|\hat{u}(x_{t_2}, x_{t_3}) - u(x_{t_2}, x_{t_3})| < 0.01\delta_l\epsilon_l$. Hence we showed that $\hat{\Delta}(x_t) > 2\epsilon_l$ only if $\Delta(x_t) \geq \epsilon_l$. This ensures that $\sum_{t=1}^{H} \mathbb{1}\left\{\Delta(x_t) \geq \epsilon_l \cup \hat{\Delta}(x_t) \geq 2\epsilon_l\right\} = 2$.

**Case II:** $\sum_{t=1}^{H} \mathbb{1}\{\Delta(x_t) \geq \epsilon_l\} \leq 1$. Here, we show that there should not exist any two $t_1, t_2$ such that $\hat{\Delta}(x_{t_1}), \hat{\Delta}(x_{t_2}) \geq 2\epsilon_l$ and at the same time $\Delta(x_{t_1}), \Delta(x_{t_2}) < \epsilon_l$. If this is the case, then we immediately get $\sum_{t=1}^{H} \mathbb{1}\left\{\Delta(x_t) \geq \epsilon_l \cup \hat{\Delta}(x_t) \geq 2\epsilon_l\right\} \leq 2$.

Now we show that there exist no such pair of $t_1, t_2$. We first consider when $\delta_l = \epsilon_l$. In this case, we have

$$|\hat{u}(x_{t_1}, x_{t_2}) - u(x_{t_1}, x_{t_2})| \geq 3\epsilon_l^2 = 3\delta_l\epsilon_l,$$

which is a contradiction to equation (45).

In the other case $\delta_l = \max(\delta, \epsilon_l) > \epsilon_l$, note that this is equivalent to $\sum_{t=1}^{H} \mathbb{1}\{\Delta(x_t) > 0\} \leq 1$. We show that whenever $\Delta(x_1) = 0$, then the LP (39) returns a solution such that $\hat{\Delta}(x_1) \leq \epsilon_0^2$. Note that

whenever $\Delta(x_1) = 0$, then for any $x_2$, we have $u(x_1, x_2) = \Delta(x_1)\Delta(x_2) = 0$. This implies, with high probability, for any $x_2$ we have

$$c_u(x_1, x_2) < 0 < c_l(x_1, x_2).$$

Hence there is no lower bound constraint (37) for $\hat{l}(x_1) + \hat{l}(x_2)$ as $c_u(x_1, x_2) \cdot c_l(x_1, x_2) < 0$, and $\hat{l}(x_1)$ can always take the minimum possible value from (38), which gives $\hat{l}(x_1) = 10 \log(\epsilon_0)$. Hence by the minimizing objective in (39), we get $\hat{\Delta}(x_1) \leq \epsilon_0^2$. This proves that whenever $\hat{\Delta}(x) \geq 2\epsilon_l$, we must have $\Delta(x) > 0$, which proves $\sum_{t=1}^{H} \mathbb{1}\left\{\Delta(s_t, a_t) \geq \epsilon_l \cup \hat{\Delta}(s_t, a_t) \geq 2\epsilon_l\right\} \leq 2$.

### D.4 Proof of Lemma A.2

For the true model $\mathcal{M}$ and $\mathcal{M}^1$, we can see that

$$\begin{aligned}
|R_m^1(r|x) - R_m(r|x)| &= |\epsilon_l - (R_+(r|x) - R_m(r|x)) - 2\epsilon_l R_+(r|x)|, \\
&\leq \epsilon_l |1 - 2R_+(r|x)| + 2\epsilon_l \leq 3\epsilon_l, && \text{if } \Delta(x) < 2\epsilon_l, \\
|R_m^1(r|x) - R_m(r|x)| &= |\epsilon_l - 2\epsilon_l R_m(r|x)| \leq 3\epsilon_l, && \text{if } \Delta(x) \geq 2\epsilon_l,
\end{aligned}$$

For the empirical model $\hat{\mathcal{M}}$ and $\mathcal{M}^2$, we first note that

$$|\hat{R}_+(r|x) - R_+(r|x)| \leq \sqrt{\iota_2/n(x)} \leq \sqrt{\iota_2/n_l} \leq 0.01\delta_l \epsilon_l.$$

for all $x : \exists (x_1, x_2) \in \mathcal{X}_l$, s.t. $x = x_1$ or $x_2$. Then, we can check that

$$\begin{aligned}
|R_m^2(r|x) - \hat{R}_m(r|x)| &= |\epsilon_l - (R_+(r|x) - \hat{R}_m(r|x)) - 2\epsilon_l R_+(r|x)|, \\
&\leq \epsilon_l |1 - 2R_+(r|x)| + |R_+(r|x) - \hat{R}_+(r|x)| + |\hat{R}_+(r|x) - \hat{R}_m(r|x)| \\
&\leq 4\epsilon_l,
\end{aligned}$$

if $\Delta(x) < 2\epsilon_l$, and for the else case, for both $m = 1, 2$ we have

$$|R_m^2(r|x) - R_m(r|x)| = \epsilon_l |1 - 2R_+(r|x)| + |R_+(r|x) - \hat{R}_+(r|x)| \leq 3\epsilon_l.$$

### D.5 Proof of Lemma B.1

This is a special case of Lemma B.3 in [36] with same transition and initial probability parameters for all contexts, as RM-MDP can be also considered as a special case of LMDP. To make the paper self-contained, we reproduce the proof. For any $t \in [H]$ and $m \in [2]$, we start with

$$\begin{aligned}
\sum_{(s,a,r)_{1:t}} |\mathbb{P}_m^{1,\pi} - \hat{\mathbb{P}}_m^{\pi}|((s,a,r)_{1:t}) &= \sum_{(s,a,r)_{1:t-1},s_t} |\mathbb{P}_m^{1,\pi} - \hat{\mathbb{P}}_m^{\pi}|((s,a,r)_{1,t-1}, s_t) \\
&\leq \sum_{(s,a,r)_{1:t-1},s_t} |\mathbb{P}_m^{1,\pi} - \hat{\mathbb{P}}_m^{\pi}|((s,a,r)_{1:t-1})\hat{T}(s_t|s_{t-1}, a_{t-1}) \\
&\quad + \sum_{(s,a,r)_{1:t-1},s_t} \mathbb{P}_m^{1,\pi}((s,a,r)_{1:t-1})|T - \hat{T}|(s_t|s_{t-1}, a_{t-1}) \\
&\leq \sum_{(s,a,r)_{1:t-1}} |\mathbb{P}_m^{1,\pi} - \hat{\mathbb{P}}_m^{\pi}|((s,a,r)_{1:t-1}) \\
&\quad + \sum_{(s,a,r)_{1:t-1}} \|(T - \hat{T})(s_t|s_{t-1}, a_{t-1})\|_1 \mathbb{P}_m^{1,\pi}((s,a,r)_{1:t-1}),
\end{aligned}$$

where in the first equality follows since we use the same reward models for $\mathcal{M}^1$ and $\hat{\mathcal{M}}$. Recursively applying the same argument, we get

$$\begin{aligned}
&\sum_{(s,a,r)_{1:H}} |\mathbb{P}_m^{1,\pi} - \hat{\mathbb{P}}_m^{\pi}|((s,a,r)_{1:H}) \\
&\qquad \leq \|(\nu - \hat{\nu})(s_1)\|_1 + \sum_{t=2}^{H} \sum_{(s,a,r)_{1:t-1}} \|(T - \hat{T})(s_t|s_{t-1}, a_{t-1})\|_1 \mathbb{P}_m^{1,\pi}((s,a,r)_{1:t-1}).
\end{aligned}$$

Finally, we observe that

$$\sum_{(s,a,r)_{1:H}} |\mathbb{P}^{1,\pi} - \hat{\mathbb{P}}^\pi|((s,a,r)_{1:H}) = \frac{1}{2} \sum_{m=1}^{2} \sum_{(s,a,r)_{1:H}} |\mathbb{P}_m^{1,\pi} - \hat{\mathbb{P}}_m^\pi|((s,a,r)_{1:H})$$

$$\leq \|(\nu - \hat{\nu})(s_1)\|_1 + \sum_{t=2}^{H} \sum_{(s,a,r)_{1:t-1}} \|(T - \hat{T})(s_t|s_{t-1}, a_{t-1})\|_1 \mathbb{P}^{1,\pi}((s,a,r)_{1:t-1})$$

$$= \|(\nu - \hat{\nu})(s_1)\|_1 + \mathbb{E}^{1,\pi}\left[\|(T - \hat{T})(s_t|s_{t-1}, a_{t-1})\|_1\right],$$

which concludes the lemma.

## Appendix E    Towards Non-Uniform Mixing Weights

In this appendix, we discuss the high-level idea of how to handle non-uniform mixing weights $w_1 = w, w_2 = 1 - w$. For a simplified discussion, let us assume $w_1, w_2 = \Omega(1)$, $\delta = \Omega(1)$ and the true $T, \nu$ are known in advance. We first recheck several quantities to check what becomes different. First, now the average reward is $p_+(x) := w \cdot p_1(x) + (1-w) \cdot p_2(x)$. Let differences in rewards be $p_-(x) := p_1(x) - p_2(x)$. For any $x_i, x_j \in \mathcal{S} \times \mathcal{A}$, the reward correlation function now becomes (compare this to (1)):

$$\mu(x_i, x_j) = w \cdot p_1(x_i)p_1(x_j) + (1-w) \cdot p_2(x_i)p_2(x_j). \tag{47}$$

If we construct a matrix $B \in \mathbb{R}^{SA \times SA}$ indexed by state-actions such that at its $(i,j)$ entry is given by:

$$B_{i,j} = \mu(x_i, x_j) - p_+(x_i)p_+(x_j),$$

then we get $B = w(1-w) \cdot qq^\top$ where $q$ is a vector indexed by $x$ such that $q_i = p_-(x_i)$. Note that the only difference from $w = 1/2$ case is the overall re-scaling factor $w(1-w)$. This suggests that we can still extract the same information on reward differences (compare this to (6))

$$u(x_i, x_j) := w(1-w) \cdot p_-(x_i)p_-(x_j) = \mu(x_i, x_j) - p_+(x_i)p_+(x_j).$$

This means the overall algorithm remains the same: we can still first get correlations from pure-exploration (Algorithm 2), and then recovers magnitude of $|p_-(x_i)|$ (Algorithm 4).

What complicates the problem is the recovery of signs: after solving a 2-SAT problem to find all signs of $\hat{p}_-(x)$ that satisfy (40), still there remains an ambiguity whether we have $sign(\hat{p}_-(x)) = sign(p_-(x))$ or $sign(\hat{p}_-(x)) = -sign(p_-(x))$ (with pair-wise consistency to satisfy (40)), because either $sign(p_-(x))$ or $-sign(p_-(x))$ is a consistent solution for (40). This is not a problem when the model is symmetric in signs of $p_-(x)$ (recall the symmetry argument in Appendix A.4.1), but when the prior is non-uniform, then we also need to find the exact signs of $p_-(x)$. To see this, observe that

$$p_1(x) = p_+(x) + (1-w) \cdot p_-(x), \quad p_2(x) = p_+(x) - w \cdot p_-(x),$$

and that changing the sign of $p_-(x)$ may result in a different set of pairs $(p_1(x), p_2(x))$.

Now we elaborate how to resolve the sign-ambiguity issue if $w \neq 1/2$. Let us consider several disjoint subsets of state-actions $G_1, G_2, ..., G_q \subseteq \mathcal{S} \times \mathcal{A}$ such that the following holds: (a) $\cup_{l=1}^{q} G_l = \{x \in \mathcal{S} \times \mathcal{A} : |p_-(x)| \geq \delta\}$, and (b) any $x_i, x_j \in \mathcal{S} \times \mathcal{A}$ belongs to the same $G_l$ for some $l \in [q]$ if one decides $sign(x_i)$ then $sign(x_j)$ is automatically decided. One can construct $\{G_l\}_{l=1}^{q}$ when we formulate the sign assignment problem to a 2-SAT problem. Now, if $q = O(1)$, then there will be only a small number of possible candidate empirical models that satisfy (40). If that is the case, we can compute the optimal policy for each candidate, run each optimal policy on the real environment, and take the one with more (estimated) long-term rewards.

In general, $q$ can be at most $O(SA)$, and thus creating too many candidate models from the combination of consistent sign assignments. In such case, we can extract information from third order correlations. Observe that, for some $(x_i, x_j, x_k) \in (\mathcal{S} \times \mathcal{A})^{\otimes 3}$, we define:

$$\mu(x_i, x_j, x_k) := \mathbb{E}[r_i r_j r_k | x_i, x_j, x_k] = w \cdot p_1(x_i)p_1(x_j)p_1(x_k) + (1-w) \cdot p_2(x_i)p_2(x_j)p_2(x_k).$$

Some algebra shows that from this, we can extract a quantity

$$u(x_i, x_j, x_k) := \mu(x_i, x_j, x_k) - p_+(x_i)p_+(x_j)p_+(x_k)$$
$$- (p_+(x_i)u(x_j, x_k) + p_+(x_j)u(x_i, x_k) + p_+(x_k)u(x_i, x_j))$$
$$= w(1-w)(1-2w) \cdot p_-(x_i)p_-(x_j)p_-(x_k).$$

Suppose that all three state-actions belong to the same group, *i.e.*, $x_i, x_j, x_k \in G_l$ for some $l \in [q]$. If $|1 - 2w|$ is not too small (say, larger than $\epsilon/H^2$), then by looking at the sign of $u(x_i, x_j, x_k)$, we can decide the signs of not only $x_i, x_j, x_k$, but also all elements in $G_l$.

Now note that, for each $l \in [q]$, there are only two possible assignments of signs for all elements in $G_l$. Pick any element $x_l \in G_l$. Non-negligible $u(x_i, x_j, x_k)$ suggests that we can perform a hypothesis-testing whether $sign(x_l) = +1$ or $-1$, because joint-probability of $(r(x_i), r(x_j), r(x_k))$ for any $x_i, x_j, x_k \in G_l$, for each case are at least apart by $|w(1-w)(1-2w)\delta^3|$ in total-variation distance. Thus if we can collect $O(|w(1-w)(1-2w)\delta^3|^{-2})$ number of samples of any third-order correlations from $G_l$, we can decide $sign(x_l)$. While we do not investigate this direction further in detail, it will be of independent interest to obtain a tight sample complexity (both upper and lower bounds) for non-uniform priors with an optimal design of recovery mechanism for $sign(x_l)$. We leave it as future work.

## Appendix F    Checklist