# OpenReview forum: "Reinforcement Learning in Reward-Mixing MDPs"
_NeurIPS.cc/2021/Conference — NeurIPS 2021 Poster_

### Official Review · Reviewer_6RaD · 2021-07-16

**Rating:** 5
**Confidence:** 2

**Summary:**

The authors consider a special case of latent MDPs where there are two unknown MDPs with identical state transition dynamics but different reward functions, and they refer to this setup as reward-mixing MDPs which they also define for M models, though only the case M=2 is studied. They show a lower bound for the sample complexity that is identical to the one proven by Kwon et al. when introducing latent MDPs (but with setting M=2), but for this special case. Further, they argue that they are the firsts to prove a polynomial (in relevant quantities) upper bound without extra assumptions beyond the MDP setting (besides the identical state transition dynamics that became part of the definition of RM-MDPs). Kwon et. al. had looked at a few other special cases like distinguishable latent MDPs.


**Limitations And Societal Impact:**

ok

**Main Review:**

The authors consider a special case of latent MDPs where there are two unknown MDPs with identical state transition dynamics but different reward functions, and they refer to this setup as reward-mixing MDPs which they also define for M models, though only the case M=2 is studied. They show a lower bound for the sample complexity that is identical to the one proven by Kwon et al. when introducing latent MDPs (but with setting M=2), but for this special case. Further, they argue that they are the firsts to prove a polynomial (in relevant quantities) upper bound without extra assumptions beyond the MDP setting (besides the identical state transition dynamics that became part of the definition of RM-MDPs). Kwon et. al. had looked at a few other special cases like distinguishable latent MDPs.

The authors claim that they had to introduce several new techniques for achieving such results and indeed, unlike the work by Kwon et. al., this article does not extend familiar techniques from the MDP setting, which also makes it much harder to read this paper. I would find it easier if they tried to reconnect by discussing what becomes of the method for M=1 and what kind of bounds would be achieved for MDPs with the resulting technique.

I think what makes identical state transitions a bit easier to handle, is that they can rely on pure exploration ideas developed for handling a reward function that is chosen after such a pure exploration phase. What the authors have to handle besides such work, is that even after the pure exploration when it is time to act near-optimally, they will for each episode not know what reward function is chosen (identifying the context in the terms of Kwon et. al)  and they also have to estimate the two reward functions without ever knowing which one was in play, challenges shared with latent MDPs in general.

I find that is would be easier to read this article if it were more conceptual structuring and connect as much as possible, not just in a related works section, to the existing work on MDPs and latent MDPs. It is interesting if this special case of latent MDPs comes with the possibility of desirable sample complexity bounds, though I have not succeeded in properly evaluating the proofs of this.


**Time Spent Reviewing:**

3-4 hours

---

> ### Author Response · Authors · 2021-08-10
> **Response to Reviewer 6RaD**
>
> We appreciate your time and effort for the review. We will work our best to deliver our message more concretely and clearly by contrasting to the $M = 1$ case as suggested.
>
> **Comparing to Kwon et al.:** While RM-MDP is a special case of LMDP, the main focus of this work is to tackle the problem without any additional information on hidden contexts. In contrast, Kwon et al. mainly considered scenarios where the hidden context information is given, or the hidden context can be inferred after the end of each episode. Absence of such information necessitates the collection of higher-order moments, which we believe is a substantial difference (not only in algorithms but also in the analysis) and opens up new avenues for future research.

---

### Official Review · Reviewer_EDyq · 2021-07-17

**Rating:** 7
**Confidence:** 2

**Summary:**

The paper proposes an algorithm to learn __reward-mixing MDPs__ in polynomial time. Reward mixing MDPs (RM-MDPs) are finite-horizon MDPs where, in each episode, the reward function is selected randomly from a set of possible reward functions. In this sense, RM-MDPs can be seen as a particular case of a POMDP where part of the state (the current reward function) is not observable.

The paper considers the case where the set of rewards contains only 2 binary reward functions, selected with uniform probability, and proposes a PAC algorithm to learn a near-optimal policy for RM-MDPs with large probability. To do so, the paper defines two quantities, $p_+$ and $p_-$ that, for each action pair $(s, a)$ correspond to the average reward ($p_+$) and the average difference of rewards ($p_-$). The individual rewards can be recovered from $p_+$ and $p_-$, which, in turn, can be computed by building an augmented MDP in which the states correspond to pairs of states in the original MDPs, and the rewards are the product of the corresponding rewards. The reward function for this augmented MDP can be efficiently estimated using, for example, reward-free exploration and then used to compute $p_+$ and $p_-$ through linear programming. Once $p_+$ and $p_-$ are computed, the original rewards can be recovered, and efficient planning can be used to derive a near-optimal policy.

The paper describes the algorithm and its theoretical analysis in terms of the number of episodes required to learn a near-optimal policy.

**Ethical Concerns:**

None.

**Limitations And Societal Impact:**

The paper's contributions are theoretical and relatively abstract in nature, so a discussion of societal impact is not very pertinent, in my opinion.

**Main Review:**

I start by saying that although I am mostly unfamiliar with the line of work pursued in this paper---aligned with PAC analysis of RL algorithms and efficient exploration. For this reason, I am unfamiliar with key pieces of relevant literature and the methods of analysis. For this reason, I have difficulty in properly assessing the novelty and technical soundness of the paper.

This said, overall, I liked the paper. The proposed motivation convinced me, and the overall methodology proposed in the paper strikes me as technically grounded and novel. I have several questions/observations that I hope may help to clarify the paper further.

1. My first observation is that the paper is notationally heavy, and the notation selected is not always self-evident. For example, it took me some time to properly understand the purpose of the augmented MDP and how it meets the needs of the algorithm. Although I realize that the paper is technically involved and notation-heavy, I believe that a little effort could be made to provide some more intuition along with the math.

2. If I understood correctly, each episode of the pure exploration is used to "visit" at most a single state of the augmented MDP and build the corresponding "reward estimate" $\hat{\mu}$. Wouldn't it make sense to reset $i$ once it reaches 3? This could allow (eventually) visiting more than one state in a single episode, or am I missing something?

3. If I understood correctly, the RM-MDP framework discussed in this paper is a particular case of latent MDPs (LMDPs). RM-MDP corresponds to LMDPs the transition probabilities and initial distributions for the different MDPs are the same. In that case, they require some form of separation between the MDP dynamics. How does this relate to the separation in (7)? More generally, I would like to understand better how the assumptions made in the LMDP model are instantiated in the RM-MDP framework and how they relate to the implicit assumptions in the paper.

**Time Spent Reviewing:**

3

---

> ### Author Response · Authors · 2021-08-10
> **Response to Reviewer EDyq**
>
> We appreciate your positive feedback. We will put more effort on keeping the paper concise and intuitive.
>
> **On resetting $i$:** Yes, you are exactly right - we can collect more correlations within the same episode once we reach $i = 3$. However, such restarting could at best improve the sample complexity by a factor of at most $H$. We believe it could be of interest if the main focus moves to optimizing factors on $H$ in the future.
>
> **Difference between (7) and separation in Kwon et al.:** The main difference is that in our work, we allow two contexts to behave the same at any state-actions whereas in Kwon et al., *all* state-actions must be sufficiently different such that eventually the contexts of any trajectories can be identified.

---

> > ### Comment · Reviewer_EDyq · 2021-09-04
> > **Thank you for your response**
> >
> > I thank the authors for the response to the points I have raised. I am happy with the author's responses.

---

### Official Review · Reviewer_KvwV · 2021-07-20

**Rating:** 5
**Confidence:** 2

**Summary:**

This paper studies exploration in a particular latent variable MDP setting where the reward function is random chosen from one of M-reward models at the beginning of each episode. The authors restrict themselves to the setting where M=2, and provide a polynomial time algorithm to learn a $\epsilon$-optimal policy. The authors do this by developing an algorithm that approximately estimates correlations of rewards at different state-actions entries in order to build an approximate model under which they can perform planning.

**Limitations And Societal Impact:**

I felt this authors did an above average job when it came to being upfront about the limitations of their work. My overall impression of this paper is more borderline than my score in part due to this.

**Main Review:**

Originality

This paper studies to a novel theoretical setting to the best of my knowledge, albeit an arguably narrow setting.

Quality

Overall, I found this paper to be above bar from an execution perspective. The majority of the analysis is provided in the appendix, but from a high level builds on the analysis in recent work by Kwon et al. 2021. It should be noted however that their polynomial bounds seem to be a consequence of their choice of considering the more restricted M=2 setting.

Clarity

Overall, the presentation of the paper is very good. I found no issues in terms of clarity.

Significance

The two main concerns I have with this paper (and the main reason I am hesitant to recommend acceptance) are the following. First, this work considers a particular case (M=2) of a subtype of latent MDPS (where only the reward function changes at the beginning of each episode) which are themselves a special class of POMDPs. I found this particular setting to be somewhat lacking in motivation. Second, this work makes a subtle assumption that the mixing weights are provided a priori, which the authors suggest might be solved by a discretizing scheme. This would however be clearly challenging for a larger number of components (M>2) where as the authors point out there are additional conceptual difficulties. Despite this, there is a possibility that this work could be of interest to subsets of the NeurIPS community, so I would consider raising my score if the above concerns are addressed.

**Time Spent Reviewing:**

5

---

> ### Author Response · Authors · 2021-08-10
> **Response to Reviewer KvwV**
>
> We appreciate your time and effort for the review.
>
> **Comparing to Kwon et al.:** While RM-MDP is a special case of LMDP, the main focus of this work is to tackle the problem without *any* additional information on hidden contexts. In contrast, Kwon et al. mainly considered scenarios where the hidden context information is given, or the hidden context can be inferred after the end of episode using a separation assumption. Absence of such information necessitates the collection of higher-order moments, which we believe is a big difference (not only in algorithms but also in the analysis).
>
> On $O(S^2 A^2)$-sample complexity, yes you are correct - it is a consequence of $M = 2$. It coincides with the lower bound $\Omega((SA)^M)$ provided in Kwon et al. with $M = 2$ when no assumptions are given on hidden dynamics, though we needed to provide another lower bound examples through Theorem 3.3 as RM-MDP is a special case.
>
> **Lacking motivation:** Please see our discussion on "Why only for $M = 2$?" in the comment for all the reviewers.
>
> **Subtle assumption that the mixing weights are provided a priori:** Please see our discussion on "simplifying assumptions" in the comment for all the reviewers.

---

### Official Review · Reviewer_qQka · 2021-07-22

**Rating:** 7
**Confidence:** 3

**Summary:**

This paper proposes a novel learning algorithm in a specific class of POMDPs known as reward-mixing-MDPs.  The paper blends clever representations, theoretical bounds, and a clean policy learning algorithm to help identify the unknown reward function for the special case of M=2 possible reward functions.  The paper is purely theoretical, and presents no experiments.

**Ethical Concerns:**

There are no ethical issues with this paper.

**Limitations And Societal Impact:**

This is not applicable, given the nature of the work.

**Main Review:**

I liked this paper.  While it is outside my area, and I did not understand it deeply, and while I have significant concerns about its significance, I think it makes some important contributions, and I recommend acceptance.  Specifically:

* A strength of the paper is its generality.  The paper makes very few assumptions on the nature of the RM-MDP to be solved, and yet is able to give an algorithm with polynomial complexity.

* The paper explicitly considers the (realistic) case of inexact, incomplete oracles.

* The idea of the "Second-Order MDP" is intriguing, and a helpful auxiliary construct that leads to the algorithm; it may be useful in other contexts.

* The use of LPs is clean and tidy.  Love it.

* The paper is generally well-written and understandable.  I was grateful that the authors took the time to build intuition about the algorithm and the logical flow of the bounds.

In terms of NeurIPS criteria:

* Originality: the paper seems original.
* Quality: the paper is of very high quality, with a solid logical flow, good references, and clear writing.
* Clarity: the paper is heavily theoretical, and therefore is heavy on notation.
* Significance: it is not clear how significant this paper is, mostly because the RM-MDP formalism seems fairly contrived, and the results are further limited to the case of M=2.  The paper does not give even passing examples of what sort of real-world phenomena this might actually model, so the impact of the paper will largely be on a small community of like-minded scholars.  The best hope is that it will spark additional results on more realistic model classes, but this seems somewhat unlikely given the close connection between the solution methodology and the specific assumptions of the model class.

As a minor note, I wasn't sure how useful Thm. 3.2 is, mostly because I am interested in the "other" type of bound: assuming that the RM-MDP exhibits some sort of structure, can we do much better than S^2A^2 sample complexity?



**Time Spent Reviewing:**

1

---

> ### Author Response · Authors · 2021-08-10
> **Response to Reviewer qQka**
>
>
> We are grateful for your positive comments. We will improve the readability of the paper by lightening burden on notations following your comments.
>
> **On model assumptions:** Please see our discussion on "Why only for $M=2?" in the comment for all the reviewers.
>
> **Is $O(S^2 A^2)$ upper bound tight?:** Quadratic dependency on dimension of state-action spaces emphasizes the contrast to standard MDPs where we would only need $O(SA)$ samples. We think that this is necessary (due to the lower-bound Theorem 3.3) without any side-information, e.g., {\it a priori} known context-revealing actions, revealed identity of the model in hindsight, etc.

---

### Official Review · Reviewer_TkXV · 2021-08-01

**Rating:** 6
**Confidence:** 4

**Summary:**

This paper studies episodic reinforcement learning (RL) in a finite-horizon reward-mixing Markov decision process (RM-MDP). Specifically, the authors have set up the problem in Section 2; and motivated, developed, and analyzed an algorithm (Algorithm 1,2,3) in Section 3. The main theoretical results are summarized in Theorem 3.2 (upper bound on the sample complexity) and Theorem 3.3 (lower bound on the sample complexity). The authors have provided an overview of the analysis in Section 4, and discussed potential future work in Section 5.

**Limitations And Societal Impact:**

I have discussed the limitations of this paper above.

This is a theoretical paper, and it is hard to speculate its negative societal impacts.

**Main Review:**

I become more positive after reading the rebuttal and other reviews. In particular,

1) I still think the M=2 limitation is a major issue of this paper. In particular, as the authors have discussed, I do not think the methodology developed in this paper can be easily extended to cases with M>2. However, as other reviewers have mentioned, M=2 is already a progress.

2) For the simplifying assumptions (e.g., the rewards are Bernoulli, the transition model is fixed, and the mixing weights are known), if the authors can well explain how to relax them in a response and/or in a section in the appendices, then I am fine with these assumptions.

The following is the original review.
***************************************************************************

This paper is interesting and in general well written. The idea of using higher-order moments to develop efficient RL algorithms for RM-MDPs is interesting and thought-provoking. However, I do not think the current version of this paper has met the bar of a top-tier machine learning conference, due to the reasons listed below:

1) My main concern is that the RM-MDPs considered in this paper are too restrictive. Specifically, in this paper, the authors have assumed that (1) M=2, where M is the number of reward models, (2) the rewards are Bernoulli, (3) the transition model is fixed, and (4) the mixing weights are known. These assumptions, especially the first two, seem to be very strong. Moreover, as the authors have discussed in Section 5, these assumptions are essential for the proposed algorithm to work. For instance, relaxing Assumption (1) or (2) might require a complete redesign, rather than a straightforward modification, of the proposed algorithm.

The authors have already mentioned some ideas of extending this paper to more general settings. I think this paper will be much stronger if the authors can relax Assumption (1) to general M>=2 and Assumption (2) to structured reward distributions, and propose a more general algorithm under these relaxed assumptions.

2) For Section 2 (Problem Setup), it would be great if the authors can further clarify which parts of the RM-MDP model are unknown to the agent before it interacts with the model. Also, for a history-dependent policy \pi, it only depends on the history in the current episode, but not the history before the current episode, right? This part is not clear from the paper, please clarify.

3) I think some numerical experiments can further strengthen this paper. For instance, the authors might provide a numerical experiment to show how to apply the developed algorithm in a simplified dynamic web problem mentioned in Section 1. The authors might also use numerical examples to check

    - if the upper bound developed in Theorem 3.2 has a uniform dependence on \epsilon
    - the performance of the proposed algorithm when the mixing weights or M are misspecified (e.g. when M is actually 1 but the agent thinks M=2)

**Time Spent Reviewing:**

7

---

> ### Author Response · Authors · 2021-08-10
> **Response to Reviewer TkXV**
>
>
> We highly appreciate your time and effort for the review.
>
> **Reply to Point 1:** Please see our discussion on simplifying assumption in the comment for all the reviewers.
>
> **Reply to Point 2:** At the beginning of every episode, the prior (or belief) over contexts is reset to $(1/2, 1/2)$. Thus, the policy we target to obtain only needs to consider events that happened in the same episode. We will clarify this in the revision.
>
> **Reply to Point 3:** We greatly appreciate this suggestion. We will consider incorporating experiments to improve our paper in the final version.

---

> ### Author Response · Authors · 2021-09-05
> **Response to the Update**
>
> We really appreciate the reviewer's positive update and his/her flexibility. We would like to explain our simplifying assumptions in more detail below and how they can be relaxed.
>
> **Bernoulli Rewards Assumption**: As mentioned in our previous common response, the assumption can be relaxed. We only need to verify the correctness of recovered parameters with rewards having the support size of $l$. This part is what our Lemma 4.3 guarantees - if pair-wise correlations of three (distinguishable) state-actions are well-estimated, then we can similarly verify that $|| \hat{p} (x) - p (x) ||_1 < \epsilon$ where $p (x)$ is redefined as $[(R_1 - R_2)(r=q_1|x), ..., (R_1 - R_2) (r=q_l|x)]^\top$ and $\hat{p} (x)$ being the corresponding estimated parameters from LP (with $l$-times more variables). Once this is guaranteed by some matrix-concentration analysis, that gives guarantees for singular-value and singular-vector, the rest of the analysis (Appendix A) follows similarly.
>
> **Fixed Transition Model Assumption**: We assume that the reviewer mentioned a *fixed* model referring to the true transition model $T$ (assumed to be known in Section 4). While in the main text we have not specifically mentioned how to update transition models or to analyze the mismatch in empirically estimated and true transition models, our overall result does not require $T, \mu$ to be known in advance. In Appendix B, we have completed our upper-bound analysis of Theorem 3.2 without assuming known transition and initial probabilities, but instead using $\hat{\nu}$ and $\hat{T}$ estimated during the pure-exploration phase.
>
>
> **Non-Uniform/Unknown Mixing Weights Assumption**: For a non-uniform but known mixing-priors $w_1 = w, w_2 = 1-w$ for some given $w \in [0,1]$, overall algorithm remains the same - there will be some changes in minor algebraic details and the sign-recovery part. We will explain in detail what changes and one solution to handle it in Appendix. Once we can handle any non-uniform $w$, then we can handle unknown mixing priors by trying all discretized $w$ (we do not optimize the sample-complexity for unknown $w$ here). We think that non-uniform / unknown prior problem could be of independent interest in future if we aim to get a tighter result.

---

### Author Response · Authors · 2021-08-10
**Common response: on our model assumptions**

We thank all the reviewers for their time and constructive feedback. We first would like to address concerns on the significance of our work mentioned by all reviewers.

**Why only for $M = 2$?:** While we share the sentiment that our current scope is specific to the $M = 2$ case and thus restrictive, we believe that a simplified model has always been a good starting point to reach long-term goals and shouldn't be overlooked. The importance of studying the special case $M = 2$ might be comparable to other kinds of mixture problems, such as a mixture of Gaussians [1,2], Linear regressions [3], or more closely to our work, a mixture of product distributions over discrete domains (e.g., Bernoulli [4]). For such fundamental and well-appreciated problems, even the $M = 2$ setting was studied extensively to understand various aspects of mixture problems. Furthermore, moving to the general $M > 2$ case has always been a significant challenge, and has been only partially resolved with some breakthroughs from other fields (e.g., tensor-decomposition / sum-of-squares). While our ultimate hope is to handle the most general case $M > 2$, the problem of learning in RM-MDP seems in nature no easier than the above-mentioned problems. Hence we feel that solving the case of $M = 2$ is a significant step forward.

Furthermore, up to our best knowledge, our work is the *first* result that employs uncertainty in higher-order moments to develop efficient RL algorithms. This required us to develop several new analysis techniques that significantly depart from existing analysis tools in RL.
As we focused on the $M = 2$ case second-order moments were enough, but for general $M > 2$ we conjecture that up to $M^{th}$-order moments (tensors) would be needed. The main challenge will be to develop an efficient recovery of model parameters from (uncertain) $M^{th}$-order moments, and the analysis that shows near-optimality of the policy obtained from recovered parameters. We believe that our result, even though it is currently only for $M = 2$, paves the way forward to tackle $M > 2$ challenges in future.


[1] Balakrishnan et al., Statistical guarantees for the EM algorithm: From population to sample-based analysis.

[2] Hardt and Price, Tight bounds for learning a mixture of two gaussians.

[3] Yi et al., Alternating minimization for mixed minear regressions.

[4] Freund and Mansour, Estimating a mixture of two product distributions.




**On simplifying assumptions:** We would like to address concerns on simplifying assumptions: uniform prior on mixing weights and Bernoulli reward distributions. We call them "simplifying" assumptions because, even though relaxing them indeed incurs significant additional burden on matrix algebra and sample-complexity analysis, the underlying idea / overall procedure of the algorithm does not seem to need major reform. We assumed those only to minimize distraction and to keep the presentation as concise and clean as possible.

For instance, in order to handle reward distributions over $l$ finite supports, we only need to verify Lemma 4.3, now showing $|| \hat{p}(x) - p (x) || < \epsilon$ instead of $| \Delta(x) - \hat{\Delta}(x)| < \epsilon$, where $p (x)$ is redefined as $[(R_1 - R_2)(r=q_1|x), ..., (R_1 - R_2) (r=q_l|x)]^\top$ and $\hat{p}(x)$ being the corresponding estimated parameters from LP (with $l$-times more variables). The main point would be the matrix concentration argument which would increase the sample-complexity by $poly(l)$-factors, but they do not affect the overall design of our algorithm. We will provide an additional section in the Appendix to discuss in more depth how to relax simplifying assumptions.

---

### Decision · Program_Chairs · 2021-09-27

**Decision:**

Accept (Poster)

**Comment:**

The main critique of the paper rests with the required assumptions for the theory.  Reviewer TkXV summarized them well.  The author response clarified that some of these assumptions were just out of simplicity in explication, and could relatively easily be relaxed.  The authors should absolutely make this more in clear in revisions of the paper, even if the details of the relaxations are better suited for supplemental material.  However, the sticking point is M=2.  This limitation is both substantial and with no easy path to relax it.  The authors argue that this is still a valuable step even if it's not settling everything.  The opposite perspective is that the theoretical machinery doesn't even lend itself toward the M>2 question, and so is a minor advance.  With the other restrictive assumptions addressed, I'm partial to theory moving forward in distinct (even if not large) steps, and this is indeed a distinct step.